# DISCOVERING CLUES OF SPOOFED LM WATERMARKS

## ABSTRACT

LLM watermarks stand out as a promising way to attribute ownership of LLM-generated text. One threat to watermark credibility comes from spoofing attacks, where an unauthorized third party forges the watermark, enabling it to falsely attribute arbitrary texts to a particular LLM. While recent works have demonstrated that state-of-the-art schemes are in fact vulnerable to spoofing, they lack deeper qualitative analysis of the texts produced by spoofing methods. In this work, we for the first time reveal that there are observable differences between genuine and spoofed watermark texts. Namely, we show that regardless of their underlying approach, all current learning-based spoofing methods consistently leave observable artifacts in spoofed texts, indicative of watermark forgery. We build upon these findings to propose rigorous statistical tests that reliably reveal the presence of such artifacts, effectively discovering that a watermark was spoofed. Our experimental evaluation shows high test power across all current learning-based spoofing methods, providing insights into their fundamental limitations, and suggesting a way to mitigate this threat.

## 1 INTRODUCTION

The improving abilities of large language models (LLMs) to generate human-like text at scale (Bubeck et al., 2023; Dubey et al., 2024) come with a growing risk of potential misuse. Hence, reliable detection of machine-generated text becomes increasingly important. Researchers have proposed the concept of watermarking: augmenting generated text with an imperceptible signal that can later be detected to attribute ownership of a text to a specific LLM (Kirchenbauer et al., 2023; Kuditipudi et al., 2023; Christ et al., 2024). Major LLM companies have pledged to watermark their models (Bartz & Hu, 2023), and regulators actively advocate for their use (Biden, 2023; CEU, 2024). However, recent works have demonstrated targeted attacks on watermarks that allow for removing the watermark or impersonating it (*spoofing*) (Sadasivan et al., 2023; Jovanović et al., 2024; Gu et al., 2024; Zhang et al., 2024). This implies that watermarks are not as robust as initially thought (Kirchenbauer et al., 2024; Piet et al., 2023).

**Red-green watermarks**   A well-studied class of LLM watermarking schemes are *Red-green* watermarks. At each step of the generation process, using both a private key $\xi$ and a few previous tokens (*context*), the watermark algorithm boosts a subset of *green* tokens, leaving other (*red*) tokens unchanged. Given a text, the detection first computes, using the private key $\xi$, the *color* of each token. A high proportion of green tokens in this *color sequence* indicates that the text is watermarked.

**Spoofing attacks**   In spoofing attacks, a malicious actor (*spoofer*) generates text that is detected as watermarked without knowledge of the private key $\xi$. Being able to generate spoofed text at scale poses a serious threat to the credibility of watermarks. Spoofed text can be falsely attributed to the model provider, causing reputational damage, or used as an argument to evade accountability (Zhou et al., 2024). Moreover, in the case of multi-bit watermarks that embed client IDs in generated text (Wang et al., 2024), spoofing attacks can be used to impersonate and incriminate a specific user. While increasing the context size may seem to be a simple way to counter spoofing, this is generally not recommended, as it greatly increases the ability of adversaries to remove the watermark from generated texts (Kirchenbauer et al., 2023; Zhao et al., 2024).

Current state-of-the-art learning-based spoofing techniques adhere to a common pipeline. First, the malicious actor queries the targeted model to build a dataset $\mathcal{D}$ of genuinely watermarked text. Then, either applying statistical methods (Jovanović et al., 2024), integer programming (Zhang et al., 2024),

| (1) Learning | (2) Generating | (3) Spoofing artifacts |

Figure 1: Overview of why spoofed texts contain measurable artifacts. First, in (1), the spoofer generates a dataset of $\xi$-watermarked texts from which he learns the watermark. Secondly, in (2), when generating text, the spoofer is better at sampling a green token if (and only if) the context and the sampled token are in his training data. This uncertainty introduces artifacts in the spoofed text. The genuine watermarking algorithm is consistent no matter the context and hence contains no such artifacts. Lastly, in (3), we build statistical tests for these artifacts to distinguish between spoofed and $\xi$-watermarked texts, even if their Z-scores $Z_\xi$ computed using the watermark detector are the same.

or fine-tuning on watermarked data (Gu et al., 2024), the spoofer learns how to forge the watermark and can generate watermarked text without additional queries to the original model (Step 1 in Fig. 1). In prior work, the success of spoofing was systematically measured using the rate of generated texts that were watermarked, with no qualitative analysis of spoofed texts' color sequences.

**Discovering artifacts in spoofed text**   In this work we, for the first time, initiate an in-depth study of spoofed text properties. We show that state-of-the-art learning-based spoofing attacks leave clues in the generated text that can be used to distinguish between spoofed text and text generated with the knowledge of the private key (Step 2 in Fig. 1). The high-level intuition behind these clues is that, at each step of generation, a spoofer has a chance to emit a green token only if the context and that token are present in their training data $\mathcal{D}$, previously obtained by querying the watermarked model. If the context is not in $\mathcal{D}$, the spoofer is forced to select the next token independently of its color. Leveraging such clues, we construct robust statistical tests that can effectively distinguish between spoofed text and genuine watermarked text generated with the private key (Step 3 in Fig. 1).

**Key contributions**   Our main contributions are:

- We provide the first in-depth analysis of artifacts in spoofed text, stressing common limitations of state-of-the-art LLM watermark spoofing methods on Red-green schemes (§3).

- We design rigorous statistical tests to practically distinguish between spoofed texts (produced by a learning-based method) and genuine watermarked texts (§4).

- We provide extensive validation of our test hypotheses and empirically show that our tests achieve arbitrarily high power given a long enough text (§5).

## 2   BACKGROUND

Given a sequence of tokens (*text*) from a vocabulary $\Sigma$, an autoregressive language model (LM) $\mathcal{M}$ outputs a logit vector $l$ of unnormalized next-token probabilities, used to sample the following token. *LM watermarking* is a process of embedding a signal within the generated text $\omega$ using a private key $\xi$ (often by modifying $l$ or the sampling procedure, see below and §6), such that this signal is later detectable by any party with access to $\xi$. In particular, a *watermark detector* $D_\xi \colon \Sigma^* \to \{0, 1\}$ implements a statistical test with the null hypothesis "*the given text was produced with no knowledge of $\xi$*". $D_\xi(\omega) = 1$ implies that the null hypothesis was rejected, i.e., the text $\omega$ is *watermarked*.

**Red-green watermarks**   We focus on the well-studied class of *Red-green watermarks*, introduced by Kirchenbauer et al. (2023); we review relevant follow-up work in detail in §6. Let $\omega_t \in \Sigma$ be the token generated by the LM at step $t$, $h \in \mathbb{N}$ the watermark's *context size* (we refer to $h$ previous tokens $\omega_{t-h:t-1}$ as the *context*), $\xi \in \mathbb{N}$ the watermark's private key, $H \colon \Sigma^h \to \mathbb{N}$ a hash function,

$PRF : \mathbb{N} \times \mathbb{N} \to \mathcal{P}(\Sigma)$ a pseudorandom function, and $\gamma, \delta \in \mathbb{R}$ watermark parameters. At each step $t$, $PRF$ uses the hash of the context $H(\omega_{t-h:t-1})$ and the private key $\xi$ to partition the vocabulary $\Sigma$ into two *colors*, $\gamma|\Sigma|$ *green* tokens (*greenlist*) and the remaining *red* tokens (*redlist*), where $\gamma$ is the watermark parameter. To insert the watermark, we modify the logit vector $l_t$ by increasing the logit of each green token by $\delta > 0$. While many hash functions $H$ have been proposed (Kirchenbauer et al., 2024), we focus on two variants proposed in Kirchenbauer et al. (2023): *SumHash* and *SelfHash*.

The shift by $\delta$ increases the ratio of green tokens in generated text, which is detectable by the detector. Namely, given a text $\omega \in \Sigma^T$, the watermark detector $D_\xi$ determines the number of green tokens $n_{green}$ and computes the Z-statistic $Z_\xi(\omega) = (n_{green} - \gamma T)/\sqrt{T\gamma(1-\gamma)}$, which under the null hypothesis follows a standard normal distribution. Finally, $D_\xi(\omega) = 1$ (i.e. $\omega$ is considered watermarked) if $Z_\xi(\omega) > \rho$. As in Kirchenbauer et al. (2023), we set $\rho = 4$.

**Watermark spoofing**    Recent work studies *spoofing attacks* (Sadasivan et al., 2023), whose goal is to reverse-engineer the watermark enough to be able to induce *false positives* in the watermark detector, i.e., generate watermarked texts without access to the private key $\xi$. So far, there are two approaches that generalize across Red-green schemes, have the ability to generate arbitrary amounts of diverse spoofed text in a cost-effective way, and are applicable in realistic setups: *Stealing* (Jovanović et al., 2024) and sampling-based *Distillation* (Gu et al., 2024) (see §6 for a discussion of other related work, and App. I for a broader discussion regarding the field of watermark spoofing).

Both methods query the watermarked model $\mathcal{M}$ to generate a dataset $\mathcal{D}$ of watermarked text. Stealing approximately infers the vocabulary splits by comparing frequencies of tokens in $\mathcal{D}$ (conditioned on the same context) with human-generated text, and uses this information to generate spoofed text using an auxiliary LM. In contrast, Distillation directly fine-tunes an auxiliary LM on $\mathcal{D}$, effectively distilling the watermark into the model weights. After the respective stealing/distillation procedure, both methods can generate an arbitrary number of spoofed texts with high *success rate*, i.e., fraction of spoofing attempts that result in high-quality (e.g., low perplexity) text that is detected as watermarked by $D_\xi$. Importantly, with such *learning-based* methods, it is possible to generate spoofed text with no additional queries to the watermarked model, making these methods practical.

In the following, we refer to watermarked text generated by such methods as *spoofed*, and use $\xi$-*watermarked* to refer to genuine watermarked text, produced using $\mathcal{M}$ and the private key $\xi$.

## 3    CAN SPOOFING ATTEMPTS BE DISCOVERED?

In this section, we discuss discoverability of spoofing, introduce the problem statement of distinguishing $\xi$-watermarked and spoofed texts, and formalize it within a hypothesis testing framework (§3.1). We then describe the intuition behind our approach (§3.2), that we later present in detail in §4.

### 3.1    PROBLEM STATEMENT

As previously discussed, current spoofing methods (*spoofers*) are evaluated in terms of their success rate at generating high-quality watermarked text. We aim to initiate a deeper qualitative study of spoofers, trying to get better insight into how well they mimic watermarked texts, beyond simply fooling watermark detectors. Our hypothesis is that due to the bottleneck of learning from a dataset of watermarked text of limited size, these spoofers, despite adopting fundamentally different approaches, may all leave similar artifacts in spoofed texts. In particular, we ask:

*Do learning-based spoofing techniques leave discoverable artifacts in generated texts?*

Showing existence of such artifacts would provide valuable insight into the shared limitations of current state-of-the-art watermark spoofers. Moreover, reliably identifying them would enable us to distinguish between $\xi$-watermarked and spoofed texts, lowering the effective accuracy of spoofers, without compromising other desirable properties, as is often the case when trying to specifically design watermarking schemes more resistant to spoofing (see §6).

Concretely, we assume the perspective of the *model provider* with a private key $\xi$ and a model $\mathcal{M}$. We receive a text $\omega \in \Sigma^T$ that is flagged as watermarked by our detector $D_\xi$, and aim to decide whether it was generated using our private key $\xi$, or by a spoofing method. Our threat model also

includes the case where we receive a *set of texts* from the same source, whose concatenation we denote as $\omega \in \Sigma^T$ for simplicity (see the bottom of §4.2 for details). We assume that our private key $\xi$ was not simply leaked; else, spoofed texts are hardly distinguishable from $\xi$-watermarked texts.

**Formalization** Determining whether a text $\omega$ was spoofed can be formulated within the hypothesis testing framework as follows:

$$H_0 : \text{The text } \omega \text{ is } \xi\text{-watermarked.} \quad H_1 : \text{The text } \omega \text{ is spoofed.} \tag{1}$$

We introduce the random variable $\Omega \in \Sigma^T$ and the received text $\omega \in \Sigma^T$ is a realization of $\Omega$. We note that the distribution of $\Omega$ under the null hypothesis and its distribution under the alternative hypothesis are different. Similarly, let $X \in \{0,1\}^T$ be the associated sequence of (non-i.i.d.) Bernoulli random variables, where $X_t = 1$ represents the event where the token $t$ is green, and let $x \in \{0,1\}^T$ be the observed color of $\omega$ under $D_\xi$ (realization of $X$). In this hypothesis testing framework, the challenge is to build a statistic $S(\Omega)$ that satisfies two key properties. First, the distribution of $S(\Omega)$ under the null hypothesis should be known in order to rigorously control the Type 1 error. Second, the distributions of $S(\Omega)$ under the null and $S(\Omega)$ under the alternative should be different, enabling us to distinguish spoofed and $\xi$-watermarked texts.

### 3.2 ARTIFACT: DEPENDENCE BETWEEN THE COLOR SEQUENCE AND THE CONTEXT

In this section, we explain why spoofed texts contain observable artifacts, as illustrated in Fig. 1.

**A simple example** To expand on this intuition, we start by considering an example of a perfect spoofer that produced the text $\omega \in \Sigma^T$, and knows the color of a token $\omega_t$, if and only if $\omega_{t-h:t} \in \mathcal{D}$, where $\mathcal{D}$ is the training data of the spoofer. Otherwise, if $\omega_{t-h:t} \notin \mathcal{D}$, we assume that the spoofer has chosen $\omega_t$ independently of its color. Let $I_\mathcal{D} : \Sigma^{h+1} \to \{0,1\}$ be the indicator function of the presence of a $(h+1)$-gram in $\mathcal{D}$. $I_\mathcal{D}$ can be interpreted as the knowledge the spoofer has over the vocabulary splits. From above, we can assume that for all $t \in \{h+1, \ldots, T\}$:

$$P(X_t = 1|I_\mathcal{D}(\Omega_{t-h:t}) = 1) \geq P(X_t = 1|I_\mathcal{D}(\Omega_{t-h:t}) = 0) \text{ if the text is spoofed;} \tag{2a}$$

$$P(X_t = 1|I_\mathcal{D}(\Omega_{t-h:t}) = 1) = P(X_t = 1|I_\mathcal{D}(\Omega_{t-h:t}) = 0) \text{ if the text is } \xi\text{-watermarked.} \tag{2b}$$

Eqs. (2a) and (2b) reflect that the knowledge of the vocabulary split at token $t$ helps the spoofer to color $\omega_t$ green, which is its original goal. For a $\xi$-watermarked text, the knowledge of a potential spoofer has no influence on its coloring. Hence, we may be able to use $I_\mathcal{D}$ to distinguish whether a sentence is spoofed or not. We now generalize this intuition to more realistic spoofing scenarios.

**Color sequence depends on the context distribution** In practice, learning how to spoof may require observing an $(h+1)$-gram multiple times. Moreover, spoofing techniques may, albeit not necessarily explicitly, have different levels of certainty regarding the color of a token given a context. Therefore, we generalize $I_\mathcal{D} : \Sigma^{h+1} \to [0,1]$ to be the function of the frequencies of $(h+1)$-grams in $\mathcal{D}$. We make a natural assumption that the higher the frequency of $\omega_{t-h:t}$ in $\mathcal{D}$, the more certain a spoofer is regarding the color of the token $\omega_t$. For now, we will also assume that for each token in $\xi$-watermarked text, $I_\mathcal{D}$ is independent of its observed color. For $\forall t \in \{h+1, \ldots, T\}$, we assume:

$$X_t \text{ is not independent from } I_\mathcal{D}(\Omega_{t-h:t}), \text{ if the text is spoofed;} \tag{3a}$$

$$X_t \text{ is independent from } I_\mathcal{D}(\Omega_{t-h:t}), \text{ if the text is } \xi\text{-watermarked.} \tag{3b}$$

This dependence between the color and $I_\mathcal{D}(\Omega_{t-h:t})$ results in spoofing artifacts under the alternative.

**Influence of the LM** Counterintuitively, the independence assumed in Eq. (3b) may be violated. To generate $\omega_t$, the model provider first computes the logit vector $l_t$ knowing $\omega_{<t}$. Then, it computes the greenlist defined by $PRF(H(\omega_{t-h:t-1}), \xi)$, and increases the logits of green tokens by $\delta$. Finally, it samples from the newly defined probability distribution to generate the token $\omega_t$. The greenlist itself is thus indeed independent of $I_\mathcal{D}(\Omega_{t-h:t})$. Yet, $l_t$ was originally computed using $\omega_{<t}$ due to the autoregressive property of the model $\mathcal{M}$, and hence may not be independent of $I_\mathcal{D}(\Omega_{t-h:t})$.

To illustrate this point, consider a case where the token $w_t$ is the only viable continuation of $\omega_{t-h:t-1}$, i.e., $l_t$ is low-entropy. Then, Bayes' theorem implies that $I_\mathcal{D}(\omega_{t-h:t})$ is likely to be high. On the other hand, the logit increase of $\delta$ has less influence on the sampling, as it is less likely to cause a token

other than $w_t$ to be sampled—thus, the color of $w_t$ is effectively random, i.e., $P(X_t = 1) \approx \gamma$, even for $\xi$-watermarked text. Hence, the events $P(X_t = 1) \approx \gamma$ and $I_\mathcal{D}(\Omega_{t-h:t})$ *is high*, are correlated, as they occur simultaneously in case of low entropy. We investigate this dependence pattern due to $\mathcal{M}$ and confirm it experimentally in more detail in App. C.

With this in mind, to properly control for Type 1 error, we need to design a test statistic $S$ where this dependence pattern is known or can be learned for $\xi$-watermarked texts. Moreover, to maintain power, we aim to distinguish this dependence from the dependence present in the case of spoofed text, as described above, building on intuition of Eqs. (3a) and (3b).

## 4 DESIGNING A TEST STATISTIC

We proceed to introduce our test statistic $S$, deriving fundamental results regarding its distribution under the independence assumption from Eq. (3b), and in the more general case where it may be violated (§4.1). Then, we present two concrete instantiations of $S$ and discuss their trade-offs (§4.2).

### 4.1 CONTROLLING THE DISTRIBUTION

We introduce the main results regarding the distribution of $S(\Omega)$ under the null hypothesis.

**Color-score correlation**   Let $\omega \in \Sigma^T$, sampled from $\Omega$, denote the text of length $T$ received by the model provider, $x \in \{0,1\}^T$, sampled from $X$, denote its color sequence under $D_\xi$, and $y \in [0,1]^T$ denote a sequence of *scores* for each token sampled from a sequence of $T$ random variables $Y$. We defer the construction of $Y$ to §4.2, where we will build on the intuition from §3.2. As the test statistic, we use the sample Pearson correlation coefficient between $x$ and $y$, defined as

$$S(\omega) = \frac{\sum_{t=1}^{T}(x_t - \bar{x})(y_t - \bar{y})}{\sqrt{\sum_{t=1}^{T}(x_t - \bar{x})^2 \sum_{t=1}^{T}(y_t - \bar{y})^2}}. \tag{4}$$

**Independence case**   We first study the distribution of $S(\Omega)$ under the assumption that $X_i$ and $Y_i$ are independent for all $i$, as in Eq. (3b) (we refer to this as *cross-independence* between $X$ and $Y$). From this assumption, we derive the following result:

**Lemma 4.1.** *Under the cross-independence between $X$ and $Y$, and additional technical assumptions (detailed in App. H), we have the convergence in distribution*

$$Z_S(\Omega) := \sqrt{T}S(\Omega) \xrightarrow{d} \mathcal{N}(0,1).$$

We defer the proof to App. H for brevity. Therefore, given a text $\omega$, we can compute a p-value using a two-sided Z-test on the statistic $Z_S(\omega)$, which is sampled from a standard normal distribution. We will refer to this test as the *Standard* method.

**General case**   In practice, however, the cross-independence assumption between $X$ and $Y$ does not always hold (see §3.2). We make a modeling assumption motivated by the results from the independent case. Let $\mu_\Omega := \mathbb{E}[S(\Omega)]$. Under the null hypothesis (and the practical considerations outlined below), we assume that

$$\sqrt{T}S(\Omega) \sim \mathcal{N}(\mu_\Omega, 1). \tag{5}$$

Compared to Lemma 4.1, the difference is that the normal distribution is offset by $\mu_\Omega$. This introduces a key challenge: finding a way to estimate $\mu_\Omega$. To this end, we propose to use $\omega_{\leq c}$, a prefix of $\omega$ of length $c$, to prompt our model $\mathcal{M}$ to generate a new sequence $\omega'$ of length $T' := T - c$ (which is a realization of $\Omega'$). In practice, we set $c = 25$. Given the shared prefix, we expect that $\Omega_{>c} \sim \Omega'$ and hence that $\mathbb{E}[S(\Omega_{>c})] = \mathbb{E}[S(\Omega')] = \mu_\Omega$. Then we introduce the statistic $Z_R(\Omega, \Omega')$, defined by

$$Z_R(\omega, \omega') = \frac{S(\omega_{>c}) - S(\omega')}{\sqrt{1/(T-c) + 1/T'}}. \tag{6}$$

Under the null hypothesis, we have that $Z_R(\Omega, \Omega') \sim \mathcal{N}(0,1)$, as $S(\omega_{>c})$ and $S(\omega')$ are two independent samples from a normal distribution. Therefore, in the general case, at the cost of higher

computational complexity (since we need to use the model to generate the new text), we can, as in the independent case, compute a p-value using a Z-test on the statistic $Z_R(\omega, \omega')$, which is sampled from a standard normal distribution. We later refer to this test as the *Reprompting* method. For consistency, in Reprompting experiments in §5, we use $T$ to implicitly refer to $T - c$.

## 4.2 CONCRETE INSTANTIATIONS

In this section we instantiate the score sequence $Y$ and propose practical modifications to $S$.

**Construction of the token score** We propose two instantiations of the score function $Y$: one that closely follows the intuition from §3.2, and another that aims to achieve the independence assumption from Lemma 4.1. Achieving cross-independence allows the construction of a test that does not require reprompting the model, hence reducing computational complexity.

**N-gram score** For the first instantiation, the idea is to directly approximate $I_{\mathcal{D}}$, the function of $(h + 1)$-grams frequencies in $\mathcal{D}$. As $\mathcal{D}$ is not known to the model provider, we approximate it with a text corpus $\tilde{\mathcal{D}}$. Assuming that the spoofer training distribution $\mathcal{D}$ is distributed similarly to natural language, we use as $\tilde{\mathcal{D}}$ a corpus of random human generated text. We define

$$y_t := I_{\tilde{\mathcal{D}}}(\omega_{t-h:t}). \tag{7}$$

In practice, we use C4 (Raffel et al., 2020) as $\tilde{\mathcal{D}}$. We study the influence of the choice of $\tilde{\mathcal{D}}$ in App. D. Finally, to reduce the required size of $\tilde{\mathcal{D}}$ needed to obtain a good estimate of $I_{\mathcal{D}}$, we compute the frequency of unordered $(h + 1)$-grams. Because the independence assumption from Lemma 4.1 is not met in this case (see §5.1), we use the Reprompting method with this score. We later refer to this specific score as $(h + 1)$-*gram* score.

**Unigram score** For the second instantiation, the intuition is to trade-off between cross-independence and reflecting $I_{\mathcal{D}}$. Let $f : \Sigma \to [0, 1]$ be the unigram frequency in human generated text. We define

$$y_t := f(\omega_{t-h}). \tag{8}$$

We look at the unigram frequency the furthest away from $t$ in order to make the dependence between $X$ and $Y$ negligible. Yet, we remain within the context window so that $y_t$ partially reflects the information from $I_{\mathcal{D}}(\omega_{t-h:t})$ and hence still allows distinguishing spoofed and $\xi$-watermarked texts. We see in §5.1 that the cross-independence assumption is satisfied for SumHash with $h = 3$. Hence, in settings where the cross-independence is verified, we use this score with the Standard method. We later refer to this specific score as *unigram* score.

**Practical considerations** In practice, we add modifications to the statistic $S$. First, as suggested in Kirchenbauer et al. (2023), we ignore repeated $h$-grams in the sequence $\omega$. This is required to enforce the independence assumption within $X$ and the independence within $Y$. Second, to limit the influence of outliers on the score, we use the Spearman rank correlation instead of the Pearson correlation and further apply a Fisher transformation. This means that in Eq. (19), $x$ and $y$ are respectively replaced by $R(x)$ and $R(y)$, where $R$ is the rank function. Hence, the statistic used in practice is defined as

$$S(\omega) = \operatorname{arctanh}\left( \frac{\sum_{t=1}^{T}(R(x)_t - \overline{R(x)})(R(y)_t - \overline{R(y)})}{\sqrt{\sum_{t=1}^{T}(R(x)_t - \overline{R(x)})^2 \sum_{t=1}^{T}(R(y)_t - \overline{R(y)})^2}} \right). \tag{9}$$

Therefore, we also use the variance $\sqrt{\frac{1.06}{T-3}}$ instead of $\sqrt{\frac{1}{T}}$ to reflect the influence of the rank function, as suggested in Fieller et al. (1957) for the i.i.d. case.

**Combining texts** Given a set of texts from a single source, we concatenate all its elements to create a single text of size $T$. In particular, let $n \in \mathbb{N}$ and $\omega^1, \cdots, \omega^n \in \Sigma^{T_1} \times \cdots \times \Sigma^{T_n}$ such that $T_1 + \cdots + T_n = T$ for a given $T$. For the Standard method, we set $\omega := \omega^1 \circ \cdots \circ \omega^n$. For the Reprompting method, we compute $\omega'^1, \cdots, \omega'^n$ independently enforcing $T_i' = T_i - c$ and then set $\omega' := \omega'_1 \circ \cdots \circ \omega'_n$ and define $\omega_{>c} := \omega_{>c}^1 \circ \cdots \circ \omega_{>c}^n$. We verify experimentally in App. B that the concatenation operation has no influence on the distribution of the statistic. Our experiments with large $T$ in §5 are thus generally conducted on concatenated texts.

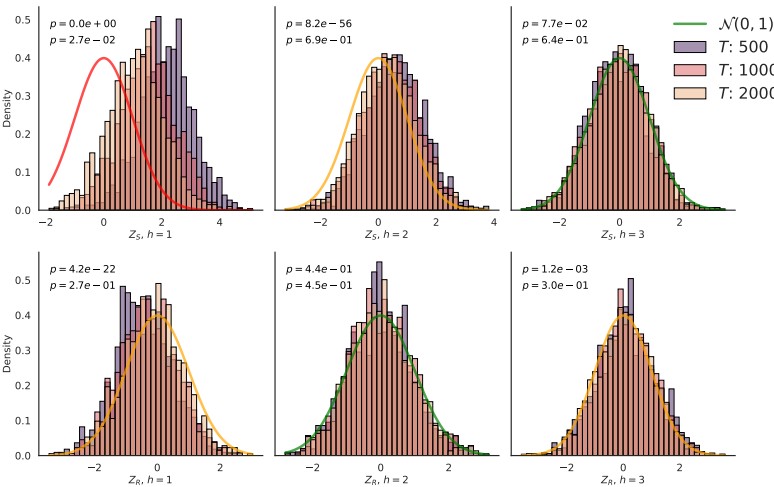

Figure 2: Histograms of $Z_S(\Omega)$ (*top*) and $Z_R(\Omega, \Omega')$ (*bottom*), with y-axes scaled to represent normalized density. The top row is computed using the unigram score and the Standard method, and the second row is computed using the $(h+1)$-gram score and the Reprompting method. A green line indicates that the $\mathcal{N}(0,1)$ hypothesis is not rejected (top p-value), an orange line that a normality test is not rejected (bottom p-value), and a red line that both are rejected at a 5% significance level.

## 5 EXPERIMENTAL EVALUATION

We present the results of our experimental evaluation. In §5.1, we validate the normality assumptions from §4.1. In §5.2, we validate the control of Type 1 error and evaluate the power of the tests from §4 on both spoofing techniques introduced in §2: *Stealing* (Jovanović et al., 2024) and *Distillation* (Gu et al., 2024). In §5.3, we compare the test results across a wider range of spoofer LMs. In App. A, we show additional results with a different watermarked model $\mathcal{M}$, parameter combinations, and another prompt dataset. In App. F, we show additional results on two additional watermarking schemes, namely AAR (Aaronson, 2023) and KTH (Kuditipudi et al., 2023), which only Distillation can spoof.

**Experimental setup**   We primarily focus on the KGW SumHash scheme, using a context size $h \in \{1,2,3\}$ and $\gamma = 0.25$. For $h \in \{1,2\}$, we set $\delta = 2$. For $h = 3$, we use $\delta = 4$ for Stealing to ensure high spoofing rates and note that Distillation is unable to reliably spoof in this setting, and therefore is excluded from our $h = 3$ experiments. In each experiment, we generate either spoofed or $\xi$-watermarked continuations of prompts sampled from the news-like C4 dataset (Raffel et al., 2020), following the methodology from prior work of Kirchenbauer et al. (2023). For each parameter combination, we generate 10,000 continuations, each being between 50 and 400 tokens long. Then, we concatenate continuations (see §4.2) to reach the targeted token length $T$. Finally, each concatenated continuation is filtered by the watermark detector, and only watermarked sequences are kept. We use those concatenated continuations to compute the test statistic $S$. In practice, we therefore have on average a total of $10^6/T$ samples per parameter combination.

We match the experimental setup from Jovanović et al. (2024) and Gu et al. (2024). In particular, we use LLAMA2-7B as the watermarked model. More specifically, in line with their original setups, we use the instruction fine-tuned version for Stealing and the completion version for Distillation. For the spoofer LM, we use MISTRAL-7B as the attacker for Stealing and PYTHIA-1.4B as the attacker for Distillation. Finally, for the spoofer training data $\mathcal{D}$, we use $\xi$-watermarked completions of C4 texts. For Stealing, $\mathcal{D}$ is composed of 30,000 samples, each 800 tokens long, whereas for Distillation, $\mathcal{D}$ is composed of 640,000 samples, each 256 tokens long. We further study the impact of $|\mathcal{D}|$ in App. E.

### 5.1 VALIDATING THE NORMALITY ASSUMPTION

In §4 we discuss two cases, each relying on one fundamental assumption. The *Independence case*: we assume independence between the color sequence $X$ and scores $Y$, from which we derive the normality of $S(\Omega)$ with a known mean (Lemma 4.1). For this case, we use the Standard method with the unigram score (Eq. (8)). The *General case*: we alternatively assume that $S(\Omega)$ is normally

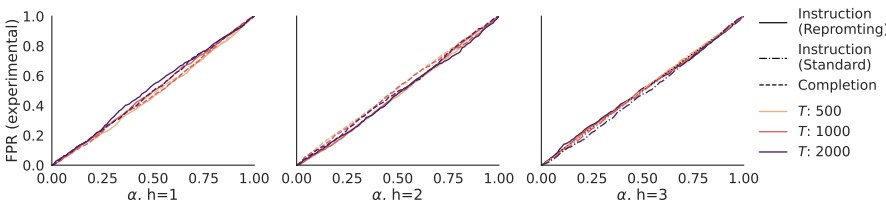

Figure 3: Experimental rejection rate of $\xi$-watermarked text on LLAMA2 7B.

distributed with an unknown mean (Eq. (5)). Here, we use the Reprompting method with the $(h+1)$-gram score (Eq. (7)). In Fig. 2, we test the Independence case assumption by validating if $Z_S(\Omega)$ with the Standard method and unigram score follows a standard normal distribution (Top), and the General case assumption by validating the same for $Z_R(\Omega, \Omega')$ with the Reprompting method and $(h+1)$-gram score (Bottom). We additionally perform both a Kolmogorov-Smirnov test for standard normality and a Pearson's normality test (not necessarily with a centered mean and unit variance).

Regarding the Independence case, we see that in the top row, $Z_S(\Omega)$ follows a standard normal distribution only for $h = 3$. This confirms our intuition behind the unigram score: as $h$ increases, the dependency between $X_t$ and $f(\Omega_{t-h})$ becomes negligible. Hence, for $h = 3$, we may use the Standard method with the unigram score, which does not require prompting our model.

For the General case, we see in the bottom row that the histogram approximately matches the standard normal distribution for $h = 2, 3$ and that the normality assumption holds for $h = 1$. Overall, these results suggest that the assumptions behind the Reprompting method are sound, allowing it (with the $(h+1)$-gram score) to be used for all tested parameter combinations. Therefore, all results in §5.2 are computed with the Reprompting method and $(h+1)$-gram score, except for $h = 3$ where both the Reprompting method with $(h+1)$-gram score and the Standard method with unigram score are used.

## 5.2 EVALUATING THE SPOOFING DETECTION TESTS

To ensure the statistical test is sound, we check whether the Type 1 error rate is properly controlled. This means that, under the null, letting $p$ be the resulting p-value, for all rejection rates $\alpha \in [0, 1]$,

$$P(p \leq \alpha) \leq \alpha. \tag{10}$$

We further evaluate the test power on Stealing and Distillation, i.e. how effective it is at distinguishing spoofed text from $\xi$-watermarked text. Additionally, we show in App. G that the Type 1 error rate remains properly controlled in the case of $\xi$-watermarked text that has been edited by humans.

**Type 1 error** To evaluate Type 1 error, we compare the experimental rejection rate under the null hypothesis against the set rejection rate $\alpha$. According to Eq. (10), if the test properly controls Type 1 error, we expect the resulting curve to be below the identity function.

In Fig. 3, we show the experimental rejection rate of $\xi$-watermarked text on LLAMA2-7B (both instruction fine-tuned and completion models) for different values of $h$ and $T$. We observe that the experimental rejection rates align closely with the identity function. Specifically, for $h = 3$, both the Reprompting method with $(h+1)$-gram score and the Standard method with unigram score align with the identity. These results show that, in practice, setting a rejection rate of $\alpha$ guarantees that the experimental False Positive Rate of the test is indeed $\alpha$.

**Test power** To evaluate the power of the test, we compute the empirical true rejection rate (i.e., TPR) under the alternative hypothesis for a given threshold $\alpha$.

In Table 1, we provide the experimental False Positive Rate (FPR, rejection under the null) and True Positive Rate (TPR, rejection under the alternative) for a fixed value of $\alpha$. For $T = 3000$, under all tested scenarios, we achieve more than 90% TPR at a rejection rate of 1%. This suggests that, given a long enough text (or concatenation of text), spoofed text from both state-of-the-art methods can be distinguished from $\xi$-watermarked text with high accuracy and reliable control over the false positive rate. Moreover, we see that the Reprompting method yields higher power than the Standard method for all values of $T$. Yet, the Standard method, in the cases where it is applicable, is computationally more efficient as it does not require prompting the model $\mathcal{M}$, and thus may still be preferable.

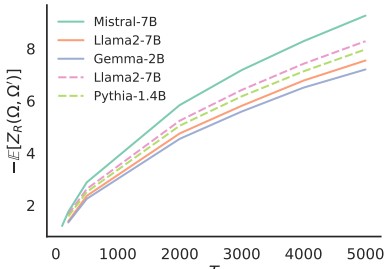

Figure 4: Experimental True Positive Rate of spoofed text. The dotted lines are the identity and serve as a reference for the expected rejection rate under the null. Since, in practice, a low false positive rate ($\alpha$) is desirable, the logarithmic scale on $\alpha$ highlights the true positive rate at low $\alpha$ values.

Table 1: Experimental FPR and TPR for both spoofers at $\alpha \in \{1\%, 5\%\}$, for different $h$ and $T$. $h = 3$ (R) denotes the Reprompting method with $(h + 1)$-gram score while $h = 3$ (S) denotes the Standard method with unigram score. All other entries are for the Reprompting method with $(h + 1)$-gram score. There are no results for Distillation with $h = 3$ as it is unable to reliably spoof in this case.

| Spoofer | | $T = 500$ | | | | $T = 1000$ | | | | $T = 2000$ | | | | $T = 3000$ | | | |
|---|---|---|---|---|---|---|---|---|---|---|---|---|---|---|---|---|---|
| | | FPR @1% | TPR @1% | FPR @5% | TPR @5% | FPR @1% | TPR @1% | FPR @5% | TPR @5% | FPR @1% | TPR @1% | FPR @5% | TPR @5% | FPR @1% | TPR @1% | FPR @5% | TPR @5% |
| STEALING | $h = 1$ | 0.00 | 0.62 | 0.04 | 0.81 | 0.00 | 0.93 | 0.04 | 0.99 | 0.01 | 1.00 | 0.05 | 1.00 | 0.01 | 1.00 | 0.07 | 1.00 |
| | $h = 2$ | 0.01 | 0.16 | 0.04 | 0.35 | 0.00 | 0.37 | 0.04 | 0.59 | 0.01 | 0.73 | 0.05 | 0.88 | 0.01 | 0.91 | 0.04 | 0.97 |
| | $h = 3$ (R) | 0.01 | 0.47 | 0.05 | 0.73 | 0.01 | 0.85 | 0.05 | 0.95 | 0.01 | 0.99 | 0.05 | 1.00 | 0.01 | 1.00 | 0.06 | 1.00 |
| | $h = 3$ (S) | 0.01 | 0.27 | 0.05 | 0.53 | 0.01 | 0.55 | 0.04 | 0.80 | 0.01 | 0.88 | 0.03 | 0.97 | 0.00 | 0.97 | 0.03 | 1.00 |
| DISTILLATION | $h = 1$ | 0.01 | 0.48 | 0.04 | 0.71 | 0.01 | 0.86 | 0.05 | 0.96 | 0.01 | 1.00 | 0.06 | 1.00 | 0.01 | 1.00 | 0.03 | 1.00 |
| | $h = 2$ | 0.01 | 0.57 | 0.06 | 0.78 | 0.01 | 0.91 | 0.06 | 0.97 | 0.01 | 1.00 | 0.05 | 1.00 | 0.00 | 1.00 | 0.07 | 1.00 |

Additionally, in Fig. 4, we show the evolution of the TPR with respect to $\alpha$. We observe that for any fixed $\alpha \in [0, 1]$, the power at $\alpha$ converges to 1 as $T$ grows. This indicates that the test can achieve arbitrary TPR at $\alpha$, given sufficiently long text. Also, we see that despite the fundamental differences between the two spoofing techniques, the texts produced by both Stealing and Distillation can be reliably distinguished with the same test. This highlights that the intuition behind our approach (§3.2) is general and that it points to a fundamental limitation of current spoofing techniques.

### 5.3 INFLUENCE OF THE SPOOFER MODEL

In this section, we run our tests on SumHash with $h = 2$, using for Stealing LLAMA2-7B, MISTRAL-7B and GEMMA 2B, and for Distillation LLAMA2-7B and PYTHIA-1.4B. Unlike §5.1 and §5.2, the results are computed with on average $10^5/T$ samples per parameter combination.

In Fig. 5, we show the evolution of the expected value of $Z_R(\Omega, \Omega')$ for spoofed texts with respect to $T$, across different spoofer LMs. We see that the evolution of the average Z-score is similar across all models, as well as for both spoofing techniques. This suggests that the choice of the spoofer LM has almost no influence on the test power.

Figure 5: Evolution of $\mathbb{E}[Z_R(\Omega, \Omega')]$ for different spoofer LMs with $T$.

Additionally, in Table 2, we show the FPR and TPR for the 5 spoofer LMs tested. For $T = 2000$, we obtain similar results across all models, with a TPR at 1% of at least 60% for Stealing and 100% for Distillation, similar to the results from §5.2. Moreover, counterintuitively, a spoofer using the same model as the model owner does not significantly lower the test power. This suggests that the artifacts we are detecting in spoofed text indeed reflect the lack of knowledge of the spoofer (§3.2), and not the difference between the LM used by the spoofer and the LM used by the model provider.

## 6 RELATED WORK

**Watermarks for LLM** In the class of distribution-modifying watermarks (Kirchenbauer et al., 2023), many schemes have built on the core idea of red-green vocabulary splits (Kirchenbauer et al., 2024; Zhao et al., 2024; Lee et al., 2023; Wu et al., 2023; Yoo et al., 2024; Fernandez et al., 2023; Liu et al., 2023; Fairoze et al., 2023; Ren et al., 2024; Lu et al., 2024; Guan et al., 2024; Zhou et al.,

Table 2: Experimental FPR at $\alpha = 1\%$ and $\alpha = 5\%$ with SumHash $h = 2$, across spoofer LMs. Bold corresponds to the case where both the spoofer and watermarked models are the same.

| Experiment | Spoofer LM | $T = 200$ | | $T = 500$ | | $T = 1000$ | | $T = 2000$ | |
|---|---|---|---|---|---|---|---|---|---|
| | | TPR @1% | TPR @5% | TPR @1% | TPR @5% | TPR @1% | TPR @5% | TPR @1% | TPR @5% |
| STEALING | **LLAMA2-7B** | 0.07 | 0.16 | 0.14 | 0.34 | 0.36 | 0.62 | 0.68 | 0.88 |
| | GEMMA-2B | 0.02 | 0.17 | 0.09 | 0.32 | 0.29 | 0.52 | 0.61 | 0.82 |
| | MISTRAL-7B | 0.05 | 0.16 | 0.16 | 0.35 | 0.37 | 0.59 | 0.73 | 0.88 |
| DISTILLATION | **LLAMA2-7B** | 0.20 | 0.46 | 0.60 | 0.80 | 0.94 | 0.99 | 1.00 | 1.00 |
| | PYTHIA-1.4B | 0.27 | 0.55 | 0.57 | 0.78 | 0.91 | 0.97 | 1.00 | 1.00 |

2024). Another prominent approach to LLM watermarking are distortion-free watermarks (Christ et al., 2024; Kuditipudi et al., 2023; Hu et al., 2024) that aim to preserve the distribution of the LM.

**Watermark spoofing**   Spoofing attacks are considered a threat to watermarks as they can lead to falsely attributing text ownership to a model provider. Sadasivan et al. (2023) presented a proof-of-concept where a dataset is generated by querying the watermarked model, and then used to approximately reverse-engineer the watermark scheme, but did not provide a practically validated method. Follow-up works Jovanović et al. (2024) and Zhang et al. (2024) expanded on this idea, to develop practical methods for spoofing Red-green watermarks. While Jovanović et al. (2024) works across multiple watermarking schemes, Zhang et al. (2024) is restricted to the unigram scheme (Zhao et al., 2024). Additionally, Gu et al. (2024) introduced an alternative spoofing method which distills the watermark into the model weights by fine-tuning a model on a dataset of $\xi$-watermarked text.

Additionally, works such as Wu & Chandrasekaran (2024) do not focus explicitly on spoofing but could be adapted to the spoofing scenario. However, in contrast with the above spoofers which can produce arbitrarily many spoofed texts once the watermark is forged, such approaches have practical limitations as they require additional queries to the watermarked model at each step of spoofing, inflating the computational cost. Moreover, another range of spoofing attacks are "piggyback spoofing attacks" (Pang et al., 2024), where an attacker substitutes a few tokens in a genuinely watermarked sentence to produce a spoofed sentence, simply using the robustness of the watermarking scheme. Piggyback spoofing attacks lack flexibility to produce arbitrary text as they rely on the targeted model for generation. Finally, there are attempts to design schemes that are more resistant to spoofing (Zhou et al., 2024), which often comes at the cost of other desirable scheme properties such as text quality.

**Broader work on LLM watermarking**   Other directions in the realm of LLM watermarking includes scrubbing attacks (Jovanović et al., 2024; Wu & Chandrasekaran, 2024; Chang et al., 2024), detection of the presence of a watermark (Tang et al., 2023; Gloaguen et al., 2024), and attempts to imprint the watermark into the model weights (Li et al., 2024; Creo & Pudasaini, 2024).

## 7   CONCLUSION

In this work, building upon the intuition that spoofed text contains artifacts reflecting the partial knowledge of the spoofer, we successfully constructed rigorous statistical tests to distinguish between spoofed and genuine watermarked texts. The tests behave similarly across the two fundamentally different spoofers studied, and across a wide range of watermark settings. Our results show that spoofed text can be reliably distinguished from genuine watermarked text, with arbitrary accuracy given a long enough text, and highlight shared limitations of current learning-based spoofers.

**Limitations**   While we can provide an experimental evaluation of power on current state-of-the-art spoofers, the proposed tests come with no theoretical guarantee of power. We build our tests on reasonable assumptions regarding the limitations of learning-based spoofing techniques. Yet, we hypothesize that spoofing techniques that adaptively learn the vocabulary split may avoid leaving similar artifacts in generated text. Designing such attacks can be an interesting path for future work. Additionally, to have high power, our tests require that the total length of the input texts is not too small. Future work could try to improve the efficiency of our method from this perspective.

## REPRODUCIBILITY STATEMENT

All technical details needed to reproduce the experiments are given in §3.2. Furthermore, each experiment both in §5 and App. A–C is introduced in detail with all parameters explicitly provided to ensure reproducibility. The theoretical result from §4.1, Lemma 4.1, is proven in App. H, where we also recall the main statistical results (Theorem H.1 and Theorem H.2) that are used in the proof.

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

Table 3: Experimental FPR and TPR for Stealing and Distillation using Dolly instead of C4 as the basis for the generation of $\omega$. The row $h = 3$ (R) corresponds to the Reprompting method with $(h + 1)$-gram score whereas $h = 3$ (S) corresponds to the Standard method with unigram score. Else only the Reprompting method with $(h + 1)$-gram score is used.

| Spoofer | | $T = 200$ | | | | $T = 500$ | | | | $T = 1000$ | | | |
|---|---|---|---|---|---|---|---|---|---|---|---|---|---|
| | | FPR @1% | TPR @1% | FPR @5% | TPR @5% | FPR @1% | TPR @1% | FPR @5% | TPR @5% | FPR @1% | TPR @1% | FPR @5% | TPR @5% |
| STEALING | $h = 1$ | 0.00 | 0.12 | 0.00 | 0.28 | 0.00 | 0.41 | 0.02 | 0.67 | 0.00 | 0.80 | 0.01 | 0.92 |
| | $h = 2$ | 0.00 | 0.03 | 0.04 | 0.15 | 0.00 | 0.12 | 0.04 | 0.28 | 0.01 | 0.34 | 0.08 | 0.50 |
| | $h = 3$ (R) | 0.01 | 0.25 | 0.03 | 0.42 | 0.02 | 0.51 | 0.05 | 0.81 | 0.01 | 0.87 | 0.05 | 0.97 |
| | $h = 3$ (S) | 0.00 | 0.16 | 0.02 | 0.43 | 0.00 | 0.27 | 0.02 | 0.51 | 0.01 | 0.48 | 0.04 | 0.75 |
| DISTILLATION | $h = 1$ | 0.01 | 0.14 | 0.05 | 0.33 | 0.00 | 0.42 | 0.02 | 0.64 | 0.00 | 0.69 | 0.02 | 0.83 |
| | $h = 2$ | 0.02 | 0.12 | 0.07 | 0.27 | 0.01 | 0.36 | 0.07 | 0.59 | 0.02 | 0.67 | 0.07 | 0.84 |

Table 4: Experimental Rejection Rate (RR) for Stealing with SelfHash and $h = 3$ for both $\xi$-watermarked text and spoofed text.

| Experiment | Method | Spoofer LM | $T = 200$ | | $T = 500$ | | $T = 1000$ | | $T = 2000$ | |
|---|---|---|---|---|---|---|---|---|---|---|
| | | | RR @1% | RR @5% | RR @1% | RR @5% | RR @1% | RR @5% | RR @1% | RR @5% |
| $\xi$-watermarked | Reprompting | / | 0.01 | 0.04 | 0.00 | 0.03 | 0.00 | 0.01 | 0.00 | 0.03 |
| | Standard | / | 0.00 | 0.03 | 0.00 | 0.02 | 0.01 | 0.02 | 0.00 | 0.01 |
| STEALING | Reprompting | LLAMA2-7B | 0.12 | 0.30 | 0.31 | 0.59 | 0.70 | 0.90 | 0.99 | 1.00 |
| | | GEMMA-2B | 0.14 | 0.30 | 0.45 | 0.73 | 0.83 | 0.93 | 1.00 | 1.00 |
| | | MISTRAL-7B | 0.10 | 0.29 | 0.38 | 0.63 | 0.79 | 0.93 | 1.00 | 1.00 |
| | Standard | LLAMA2-7B | 0.03 | 0.13 | 0.06 | 0.20 | 0.11 | 0.35 | 0.32 | 0.63 |
| | | GEMMA-2B | 0.03 | 0.14 | 0.07 | 0.26 | 0.15 | 0.40 | 0.36 | 0.62 |
| | | MISTRAL-7B | 0.05 | 0.22 | 0.15 | 0.39 | 0.35 | 0.63 | 0.74 | 0.88 |

# A ADDITIONAL EXPERIMENTAL RESULTS

In this section, we conduct several thorough ablation studies. We evaluate the test using a different dataset as base prompts (App. A.1), with a different variation of the watermark scheme (App. A.2), and using another watermarked model (App. A.3). In all tested additional settings, the results are similar to those presented in §5, which emphasizes the validity of the test and shows that the spoofing artifacts studied are a fundamental property of learning-based spoofers.

Unlike in §5, we generate 1,000 continuations per parameter combination for the ablation study. It means that on average we have $10^5/T$ samples per parameter combination.

## A.1 MITIGATING POTENTIAL METHODOLOGICAL BIASES

Here, we use the same settings as §5 (Stealing and Distillation with SumHash, different values of $h$, and for $h = 3$, both the Reprompting and Standard methods), but use text continuations of prompts sampled from Dolly (Conover et al., 2023) instead of the C4 dataset. We show that the methodology used to generate the spoofed and $\xi$-watermarked texts has no influence on the results.

In Table 3, we show the experimental FPR and TPR at $\alpha$ of 1% and 5%. The results are similar to those on C4 from Table 1: the Type 1 error is controlled, and the power is similar. This suggests that the methodology we use to generate the prompts does not influence the results. Hence, we can expect that for most texts $\omega$, the empirical results presented hold, and that if $\omega$ is spoofed, the spoofer's artifacts remain present and discoverable.

Table 5: Experimental Rejection Rate (RR) for Stealing with MISTRAL7B as $\mathcal{M}$ at $\alpha$ of 1% and 5% on both $\xi$-watermarked text and spoofed text.

| Experiment | Method | Spoofer LM | $T = 200$ RR @1% | $T = 200$ RR @5% | $T = 500$ RR @1% | $T = 500$ RR @5% | $T = 1000$ RR @1% | $T = 1000$ RR @5% | $T = 2000$ RR @1% | $T = 2000$ RR @5% |
|---|---|---|---|---|---|---|---|---|---|---|
| $\xi$-watermarked | Reprompting | / | 0.02 | 0.05 | 0.03 | 0.08 | 0.00 | 0.04 | 0.00 | 0.02 |
|  | Standard | / | 0.01 | 0.04 | 0.01 | 0.02 | 0.00 | 0.02 | 0.00 | 0.02 |
| STEALING | Reprompting | LLAMA2-7B | 0.45 | 0.73 | 0.89 | 1.00 | 0.99 | 1.00 | 1.00 | 1.00 |
|  |  | GEMMA-2B | 0.48 | 0.90 | 0.97 | 1.00 | 1.00 | 1.00 | 1.00 | 1.00 |
|  |  | MISTRAL-7B | 0.59 | 0.81 | 0.97 | 1.00 | 1.00 | 1.00 | 1.00 | 1.00 |
|  | Standard | LLAMA2-7B | 0.19 | 0.41 | 0.25 | 0.60 | 0.45 | 0.79 | 0.83 | 0.95 |
|  |  | GEMMA-2B | 0.21 | 0.48 | 0.40 | 0.66 | 0.64 | 0.81 | 0.85 | 0.96 |
|  |  | MISTRAL-7B | 0.27 | 0.55 | 0.70 | 0.88 | 0.94 | 0.99 | 0.99 | 1.00 |

## A.2 RESULTS FOR THE SELFHASH SCHEME

Next, we focus on SelfHash with $h = 3$ and $\delta = 4$ for Stealing. We use both the Reprompting and the Standard method with their respective score functions (§4.2).

In Table 4, we show the experimental FPR at $\alpha = 1\%$ and $\alpha = 5\%$ for $\xi$-watermarked and spoofed text. Similarly to the SumHash variant, the Type 1 error is properly controlled for both the Standard and the Reprompting methods. Moreover, the empirical power scaling with $T$ is similar to the SumHash scheme from Table 1. This means that the spoofing artifacts are not tied to a specific scheme, but rather represent a fundamental limitation of learning-based watermark spoofing techniques such as Stealing and Distillation. Additionally, we also see that the power of the Standard method at a fixed $T$ is lower than that of the Reprompting method. This confirms the expected trade-off of the unigram score: enforcing cross-independence is traded for power (§4.2).

## A.3 ALTERNATIVE WATERMARKED MODEL

In this experiment, we use MISTRAL-7B as the watermarked model $\mathcal{M}$ for SumHash at $h = 3$ on Stealing. We do not use a different $\mathcal{M}$ for Distillation, as Distillation was only empirically validated on LLAMA2-7B (Gu et al., 2024).

In Table 5, we show the experimental FPR at $\alpha$ of 1% and 5% for $\xi$-watermarked text and spoofed text on different spoofer LMs. Similar to the results in Table 1, the Type 1 error is controlled in both the Reprompting and Standard methods. Moreover, the power scaling with $T$ is also similar to the results from Table 1. This suggests that the model $\mathcal{M}$ used by the model provider has no influence on the artifacts left by spoofing attempts on such a model. It also confirms the results from §5.3 that the artifacts we are distinguishing in spoofed text indeed reflect only the lack of knowledge of the spoofer and do not reflect a particular behavior of a given model.

## B VALIDATING THE CONCATENATION PROCEDURE

In this section, we experimentally validate the claim that concatenating texts $\omega$ according to the procedure from §4.2 has no influence on the resulting distribution of the statistic.

**Experimental setup** Let $W = (\omega_1, \ldots, \omega_n)$ be a corpus of $n$ texts of the same length, and $W'$ the corresponding corpus of Reprompting texts of the same length $T$. Let $X, X' \in \{0,1\}^{n \times T}$ be the color matrices of the corpora, and $Y, Y' \in [0,1]^{n \times T}$ be the associated $(h+1)$-gram score matrices of the corpora. For permutations $\sigma \in \mathfrak{S}_{n \times T}$, we define $\sigma(X)_{i,j} = X_{\sigma((i,j))}$ and $\sigma(Y)_{i,j} = Y_{\sigma((i,j))}$. We define $\sigma(W)$ as the *shuffled* corpus with the corresponding $\sigma(X)$ color and $\sigma(Y)$ score. Given $\sigma \in \mathfrak{S}_{n \times T}$, we test the hypothesis that shuffling has no influence on the distribution of $Z_R(W, W')$,

$$Z_R(\sigma(W), \sigma(W')) \sim Z_R(W, W'), \tag{11}$$

where $Z_R(W, W') := (Z_R(\omega_1, \omega_1'), \ldots, Z_R(\omega_n, \omega_n'))$. The shuffling operation can be interpreted as a concatenation of texts of length 1. Hence, if the shuffling has no influence, this implies that the

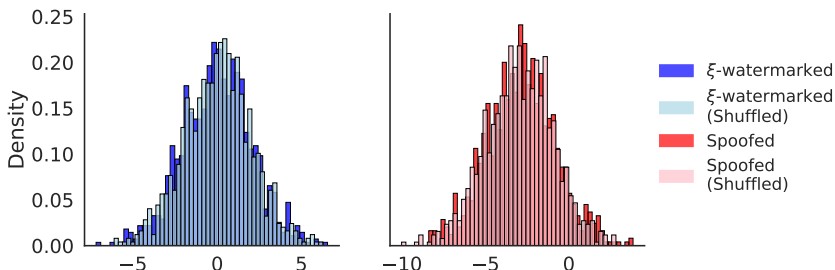

Figure 6: Histogram of Z-scores for both $\xi$-watermarked and spoofed corpora, as well as their shuffled counterparts.

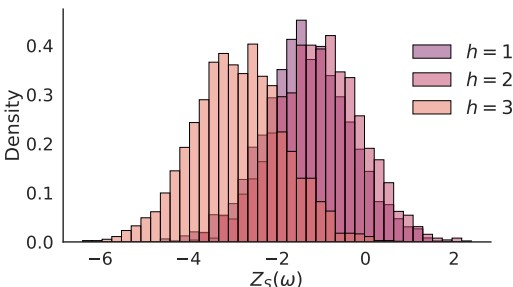

Figure 7: Histogram of $Z_S(W)$ for $\xi$-watermarked text with $(h+1)$-gram score and Standard method.

concatenation of texts of longer length has no influence either. To test for the equality of distribution, we use a Mann-Whitney U rank test.

**Results** In practice, we generate $n = 1000$ $\xi$-watermarked and spoofed texts of length 175 and their corresponding $T = 150$-length Reprompting text corpora. We sample $\sigma \in \mathfrak{S}_{n \times T}$ uniformly in $\mathfrak{S}_{n \times T}$. In Fig. 6, we show the resulting histogram of $Z_R(W, W')$ and $Z_R(\sigma(W), \sigma(W'))$. The histograms between the non-shuffled and shuffled versions perfectly overlap for both the $\xi$-watermarked texts and the spoofed texts. Moreover, the resulting p-values from the Mann-Whitney U rank test are $0.86$ and $0.44$, respectively. Hence, we can conclude that Eq. (11) is verified and that the concatenation procedure has no influence on the distribution of the statistic.

## C  DEPENDENCE BETWEEN THE CONTEXT DISTRIBUTION AND THE COLOR

In this section, we study in detail the dependence between the color of token $\omega_t$ and $I_{\mathcal{D}}(\omega_{t-h:t})$ in $\xi$-watermarked text from §3.2.

**Problem statement** We recall that $\mathcal{D}$ is the training data of the spoofer, and $I_{\mathcal{D}}$ is the function of frequencies of $h + 1$-grams in $\mathcal{D}$. In §3.2, we hypothesize that low entropy is a common factor that implies $I_{\mathcal{D}}(\Omega_{t-h:t})$ is high and $P(X_t = 1) \approx \gamma$. Under such an assumption, we therefore expect the correlation between the observed color sequence $x$ and the $(I_{\mathcal{D}}(\omega_{t-h:t}))_{\forall t \in \{h,...,T\}}$ to be negative. In other words, we expect $Z_S(\omega)$ with the $(h + 1)$-gram score to be negative for $\xi$-watermarked text.

**Results** We verify this claim by computing $Z_S(\omega)$ with the $(h + 1)$-gram score for a corpus $W$ of 1000 $\xi$-watermarked texts, each of length $T = 500$. In Fig. 7, we see the histograms of $Z_S(W)$ for different values of $h$. We see that for all $h$, $\hat{\mathbb{E}}[Z_S(W)]$ is indeed negative. Furthermore, we notice that the histograms appear normally distributed, agreeing with the assumption underlying the Reprompting method (Eq. (5)). Therefore, these results show that the proposed intuitive explanation of the dependence due to $\mathcal{M}$ is coherent, and further highlight the need for the Reprompting method in order to build a statistic with a known distribution when using the $(h + 1)$-gram score.

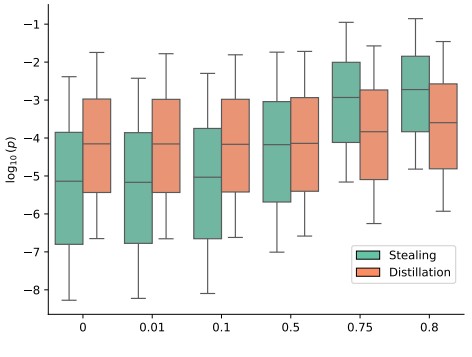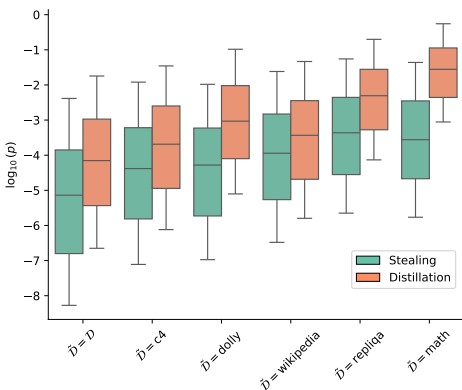

Figure 8: *Left*: Evolution of the p-value distribution with the Total Variation distance between $\mathcal{D}$ and $\tilde{\mathcal{D}}$. Note that the x-scale is not linear. *Right*: Evolution of the p-value distribution for different choices of $\tilde{\mathcal{D}}$. Each p-value is computed with 1000-token long completions. The whiskers are set at 0.5 of the IQR for visibility.

## D  INFLUENCE OF THE TRAINING DATASET

In this section, we study the influence of $\tilde{\mathcal{D}}$ on our ability to detect spoofed texts. As we do not know the true distribution of $\mathcal{D}$, we hope that using a different dataset does not significantly affect our results. We study the influence of $\tilde{\mathcal{D}}$ on both Stealing and Distillation methods with SumHash $h = 1$.

**Evolution with the TV distance**    First, we analyze the influence of the choice of $\tilde{\mathcal{D}}$ in a controlled setting. We let $\tilde{\mathcal{D}}_0$ be the counts of the different $(h + 1)$-grams in $\mathcal{D}$. We then build a perturbed dataset $\tilde{\mathcal{D}}_\epsilon$ by adding centered normal noise with standard deviation $\epsilon$ to $\tilde{\mathcal{D}}_0$. Finally, we compute the total variation distance between $\tilde{\mathcal{D}}_\epsilon$ and $\mathcal{D}$. In Fig. 8 (left), we observe that the p-values increase on average with the total variation distance between $\tilde{\mathcal{D}}_\epsilon$ and $\mathcal{D}$. This confirms the intuition that the better the estimate of $\mathcal{D}$, the more powerful our tests are. Furthermore, it appears that the p-values increase slowly with the total variation distance, which suggests that the choice of $\tilde{\mathcal{D}}$ is not crucial for obtaining a powerful test.

**Comparing different training datasets**    We run the test for different choices of $\tilde{\mathcal{D}}$ (C4 (Raffel et al., 2020), Dolly (Conover et al., 2023), Wikipedia (WikimediaFoundation), Repliqa (Monteiro et al., 2024), and Math (Fourrier et al., 2023)) as well as $\tilde{\mathcal{D}} := \mathcal{D}$ for comparison. In Fig. 8 (right), we see that even the Math dataset has reasonable p-values for Stealing despite our experimental evaluations using significantly different prompt completions from news articles. This confirms that our test is robust to the choice of $\tilde{\mathcal{D}}$.

Lastly, given a received watermarked text $\omega$, a model provider could adjust the choice of $\tilde{\mathcal{D}}$ based on the topic of $\omega$. Such a heuristic could ensure that the choice of $\tilde{\mathcal{D}}$ is always relevant, and further mitigate its impact on the test power.

## E  INFLUENCE OF THE SIZE OF THE SPOOFER TRAINING DATA

In this section, we study how the power of the test is impacted by the size of the training dataset $\mathcal{D}$.

We run the test for Stealing with LeftHash, $h = 3$, LLAMA2-7B as the watermarked model, and MISTRAL-7B as the spoofer model. We train the spoofer with different sizes of $\mathcal{D}$. We note that, for practicality, the smaller instances of $\mathcal{D}$ are subsets of the larger ones. We also set $\tilde{\mathcal{D}} = \mathcal{D}$ each time to control for the discrepancy between $\tilde{\mathcal{D}}$ and $\mathcal{D}$. Otherwise, we run the same experimental procedure as in §5.2, but using only $n = 500$ completions. Because spoofing attempts are less successful for

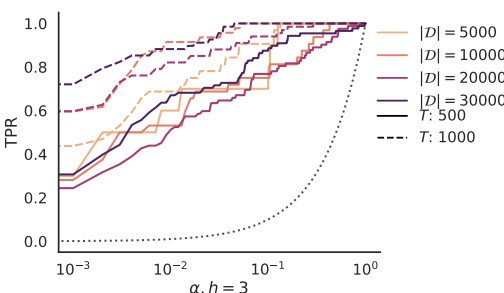

Figure 9: Experimental True Positive Rate of spoofed text with different sizes of $\mathcal{D}$. The size of $\mathcal{D}$ is measured in queries, and each query is 800 tokens long.

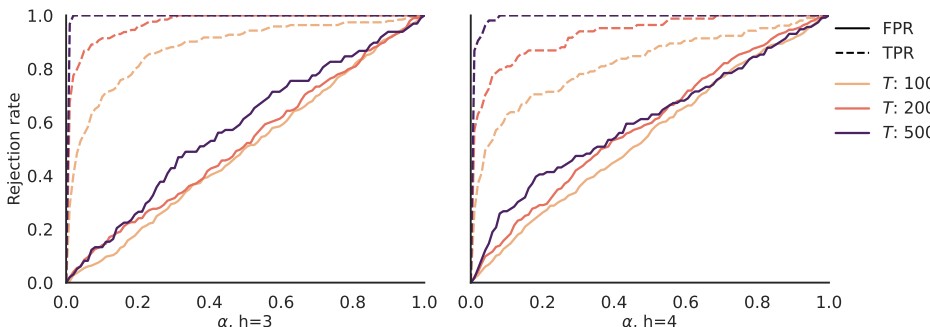

Figure 10: Rejection rates for the Reprompting method on the AAR watermark with $h = 3$ and $h = 4$. The solid lines correspond to $\xi$-watermarked text and the dashed lines to Distillation-spoofed text.

smaller $\mathcal{D}$, we have in the worst case 30 samples per parameter combination (compared to $5 \times 10^4/T$ samples per parameter combination for larger $\mathcal{D}$).

We see in Fig. 9 that there is no clear trend between the length of $\mathcal{D}$ and the power of the test, that would hold across all the tested settings. In some cases, the power of the test is lower for smaller $\mathcal{D}$—we hypothesize that one reason for this is the (necessary) filtering of failed spoofing attempts. For smaller $\mathcal{D}$, the spoofing method fails to generalize beyond its training data, and so the (rare) cases when it succeeds are the cases where the spoofed sentence is close to the training data. Ultimately, it is important to conclude that smaller $|\mathcal{D}|$ is not strictly better for the spoofing adversary—while there may be fewer artifacts, the more important goal of the spoofer (to spoof) is often greatly sacrificed.

## F    EXTENDING THE METHOD TO OTHER SCHEMES

While we design our method to detect spoofing attempts on Red-Green schemes (Kirchenbauer et al., 2023), as these are the primary target of several spoofing works, we show that the method can be generalized to other watermarking schemes. Excluding the unigram scheme by Zhao et al. (2024), which Zhang et al. (2024) shows can be perfectly spoofed, we can study the AAR scheme from Aaronson (2023), as well as one of the KTH schemes from Kuditipudi et al. (2023), as both schemes were shown to be spoofable via Distillation (Gu et al., 2024).

**AAR watermark**    In the AAR watermark, $h$ previous tokens are hashed using a private key $\xi$ to obtain a score $r_i$ uniformly distributed in $[0, 1]$ for each token index $i$ in the vocabulary $\Sigma$. Given $p_i$, the original model probability for token index $i$, the next token is then deterministically chosen as the token $i^*$ that maximizes $r_i^{1/p_i}$.

Given a text $\omega \in \Sigma^T$, we naturally generalize Eq. (4) by defining $x \in \mathbb{R}^T$ as $x_t = -\log r_{\omega_t}$, whereas previously $x_t$ was the color of the $t$-th token. The rest of the method remains identical.

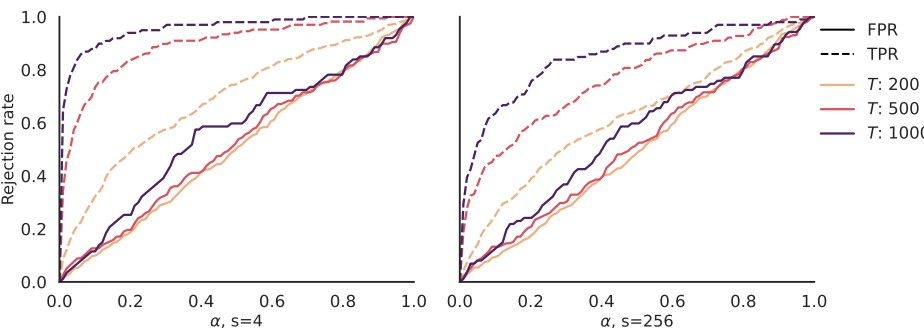

Figure 11: Rejection rates for the Reprompting method on the KTH watermark with $s \in \{1, 4, 256\}$. The solid lines correspond to $\xi$-watermarked text and the dashed lines to Distillation-spoofed text.

We evaluate both the FPR and TPR of our test using $h \in \{3, 4\}$, LLAMA2-7B as both the watermarked model and the attacker model, the Reprompting method, and the same experimental procedure as in §5, except that we generate only $n = 500$ completions. We discarded $h = 2$ as the watermarked model output was too low-quality and repetitive (Gu et al., 2024). In Fig. 10, we see that the generalized method can successfully detect spoofed text with a 90% TPR at a rejection rate of 1% for 500 tokens. In fact, it is even more powerful than the detection in the Red-Green scheme, where we achieved a similar TPR at 1% with 3000 tokens (§5.2). However, the test hypothesis appears slightly violated, as the empirical FPR at 1% is around 2% for both $h = 3$ and $h = 4$. As the variant we run is the simplest adaption of the test primarily designed for Red-Green watermarks, we believe that a more tailored test could improve this.

**KTH watermark** In the KTH watermark (EXP variant), a single watermark key sequence of length $n_{key}$, $\xi = \xi^1, \ldots, \xi^{n_{key}}$, is uniformly distributed, where each $\xi^i \in [0, 1]^{|\Sigma|}$. To generate the $j$-th token (modulo $n_{key}$), the watermark samples the token $i^*$ that maximizes $\left(\xi_i^j\right)^{1/p_i}$. Additionally, to allow more diversity in the generated text, the key is randomly shifted by a constant at each query. As in Gu et al. (2024), we denote by $s$ the number of allowed shifts.

Given a text $\omega \in \Sigma^T$, we naturally generalize Eq. (4) by defining $x \in \mathbb{R}^T$ as $x_t = \log(1 - \xi_{\omega_t}^t)$, whereas previously $x_t$ was the color of token $t$. To account for the permutation of the key, we further replace $\log(1 - \xi_{\omega_t}^t)$ with the Levenshtein cost introduced in Kuditipudi et al. (2023). Moreover, the scheme, being based on a fixed key, lacks any context $h$ that can be used to compute the N-gram score $y_t$ (Eq. (7)). Following the intuition from Gu et al. (2024) that, in the limit, their spoofing ability comes from learning contiguous watermarked sequences of length $n_{key}$, we suspect that setting $h \approx n_{key}$ would enable greater test power. In practice, due to practical constraints, we set $h = 5$. The rest of the method remains unchanged.

We evaluate both the FPR and TPR of our test, using $s = 4$, and $s = 256$, along with a key of length $n_{key} = 256$, on LLAMA2-7B as both the watermarked model and the attacker model, the Reprompting method, and the same experimental procedure as in §5, except that we generate only $n = 500$ completions. In Fig. 11, we see that this generalized method can successfully detect spoofed text for both $s = 4$ and $s = 256$, albeit with a TPR of 65% at a confidence of 99% for $s = 4$ and TPR of 30% for $s = 256$. In all three cases however, the Type 1 error is controlled, i.e., empirical FPR corresponds to the theoretical FPR. As the variant we run is the simplest adaption of the test primarily designed for Red-Green watermarks, we believe that a more tailored test could improve its power.

## G  INFLUENCE OF HUMAN MODIFICATIONS ON FPR

Here, we study the behavior of $\xi$-watermarked text that has subsequently been edited by humans. We consider two different use cases. The first is the case of cropping. Given a $\xi$-watermarked text, we assume a human inserts non-watermarked text in the middle. This corresponds to a plausible use case of LLMs, where humans merge generated text with their own. Second, we consider paraphrasing. Given a $\xi$-watermarked text, we paraphrase it using DIPPER (Krishna et al., 2023).

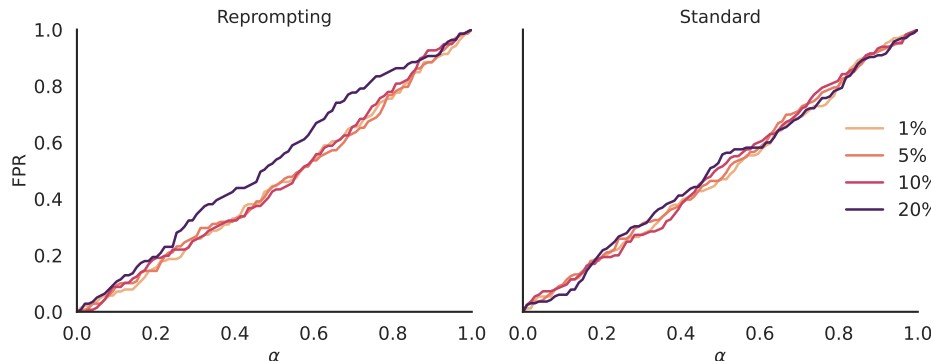

Figure 12: Experimental rejection rate of mixed $\xi$-watermarked text and human text on LLAMA2-7B for both the Reprompting method (left) and the Standard method (right) at different percentages of human text. Each mixed text is in total 500 tokens long.

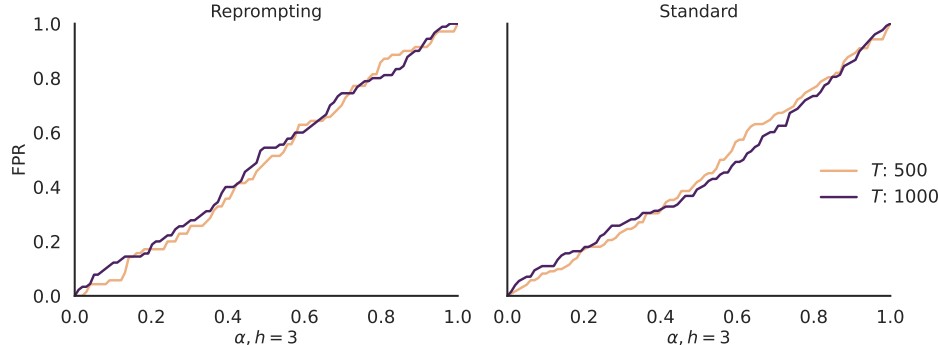

Figure 13: Experimental rejection rate of paraphrased $\xi$-watermarked text on LLAMA2-7B for both the Reprompting method (left) and the Standard method (right).

We evaluate the FPR of human-modified text using $h = 3$ on both the Standard and Reprompting methods and follow a similar experimental procedure as in §5, except that we generate only $n = 500$ samples. Given a percentage $\rho$, for each generated C4 prompt completion of length $T$, we randomly insert another random human text sampled from C4 such that $\rho$ percent of the resulting text is human-generated. We used this procedure for $\rho \in \{0.01, 0.05, 0.1, 0.2\}$. As in §5, we apply the test only on text that appears watermarked according to the original watermark detector. In Fig. 12, we see that even for the highest percentage of human text (20%), the test properly controls Type 1 error.

We evaluate the FPR of the paraphrased text using $h = 3$ on both the Standard and Reprompting methods and follow a similar experimental procedure as in §5, except that we generate only $n = 1000$ samples. We note that we apply the test only on text that is considered watermarked by the original watermark detector. In Fig. 13, we see that the test still properly controls Type I error for both methods and for different text lengths.

Both results show that a rejection rate of $\alpha$ still guarantees an experimental FPR of $\alpha$, even if the $\xi$-watermarked texts have been altered by humans.

# H   PROOF OF LEMMA 4.1

In this section, we detail the proof of Lemma 4.1.

First, let's recall some statistical results that we need.

**Theorem H.1** (Lindeberg CLT). *Let $X_{n,1}, ..., X_{n,n}$ be independent random variables in $\mathbb{R}^d$ with mean zero. If for all $\varepsilon > 0$*

$$\sum_{k=1}^{n} \mathbb{E}[||X_{n,k}||^2 \mathbb{1}\{||X_{n,k}|| > \varepsilon\}] \to 0, \ \textit{(Lindeberg Condition)} \tag{12}$$

*and*

$$\sum_{k=1}^{n} cov(X_{n,k}) \to V, \tag{13}$$

*then*

$$\sum_{k=1}^{n} X_{n,k} \xrightarrow{d} \mathcal{N}(0, V). \tag{14}$$

**Theorem H.2** (Delta method). *Let $X_1, ..., X_n$ be a sequence of random variables in $\mathbb{R}^d$, if*

$$\sqrt{n}(X_n - \mu) \xrightarrow{d} \mathcal{N}(0, V), \tag{15}$$

*and $u : \mathbb{R}^d \to \mathbb{R}$ is differentiable at $\mu$, with $\nabla u(\mu) \neq 0$, then*

$$\sqrt{n}(u(X_n) - u(\mu)) \xrightarrow{d} \mathcal{N}(0, \nabla u(\mu)^T V \nabla u(\mu)). \tag{16}$$

Now we proceed to prove Lemma 4.1. We first state the result formally.

**Lemma 4.1.** *Let $X := X_1, \ldots, X_T$ be a sequence of independent (non i.i.d) Bernoulli random variables, and $g_i = P(X_i = 1)$. Let $Y := Y_1, \ldots, Y_T$ be a sequence of i.i.d. random variables. Let $\Omega = (X, Y)$. Assuming that, for all $i \in \{0, \ldots, T\}$, $X_i$ and $Y_i$ are independent, that there exist $g^{(1)}, g^{(2)} \in [0, 1]$ such that*

$$\lim_{T \to \infty} \frac{1}{T} \sum_{i=1}^{T} (g_i - g^{(1)}) = O\left(\frac{1}{T}\right) \textit{ and } \lim_{T \to \infty} \frac{1}{T} \sum_{i=1}^{T} g_i^2 = g^{(2)}, \tag{17}$$

*and assuming that $Y$ admits at least 4 moments $\mu_Y, \mu_{Y^2}, \mu_{Y^3}, \mu_{Y^4}$. Then, we have that*

$$Z_S(\Omega) := \sqrt{T} S(\Omega) \xrightarrow{d} \mathcal{N}(0, 1). \tag{}$$

*Proof.* Let $w_i := (X_i, Y_i, X_i^2, Y_i^2, X_i Y_i)$. Let $X_{n,k} = \frac{(w_i - \mathbb{E}[w_i])}{\sqrt{n}}$. We recall the definition of $S$,

$$S(\Omega) = \frac{\sum_{t=1}^{T} (X_t - \bar{X}_T)(Y_t - \bar{Y}_T)}{\sqrt{\sum_{t=1}^{T} (X_t - \bar{X}_T)^2 \sum_{t=1}^{T} (Y_t - \bar{Y}_T)^2}}. \tag{18}$$

$$= \frac{\frac{1}{T} \sum_{t=1}^{T} X_t Y_t - \left(\frac{1}{T} \sum_{t=1}^{T} X_t\right)\left(\frac{1}{T} \sum_{t=1}^{T} Y_t\right)}{\sqrt{\frac{1}{T} \sum_{t=1}^{T} X_t^2 - \left(\frac{1}{T} \sum_{t=1}^{T} X_t\right)^2} \sqrt{\frac{1}{T} \sum_{t=1}^{T} Y_t^2 - \left(\frac{1}{T} \sum_{t=1}^{T} Y_t\right)^2}}, \tag{19}$$

where $\bar{X}_T$ denotes the mean of $X_{1:T}$.

The proof goes as follows:

- First, we show that the sum of the covariance matrix of $X_{n,k}$ converges (Eq. (13)).

- Then, we show that $X_{n,k}$ satisfies the Lindeberg condition (Eq. (12)). We can then apply the Lindeberg theorem to show that $w_i$ converges to a normal distribution.

- Finally, we apply the Delta method (Theorem H.2) to show that $S(\Omega)$ is normally distributed.

We have that for all $i \neq j$, $w_i$ is independent of $w_j$. For each $i$, we have

$$\text{Cov}(w_i) = \begin{bmatrix} g_i(1-g_i) & 0 & g_i(1-g_i) & 0 & \mu_Y g_i(1-g_i) \\ 0 & -\mu_Y^2 + \mu_{Y^2} & 0 & -\mu_Y \mu_{Y^2} + \mu_{Y^3} & g_i(-\mu_Y^2 + \mu_{Y^2}) \\ g_i(1-g_i) & 0 & g_i(1-g_i) & 0 & \mu_Y g_i(1-g_i) \\ 0 & -\mu_Y \mu_{Y^2} + \mu_{Y^3} & 0 & -(\mu_{Y^2})^2 + \mu_{Y^4} & g_i(-\mu_Y \mu_{Y^2} + \mu_{Y^3}) \\ \mu_Y g_i(1-g_i) & g_i(-\mu_Y^2 + \mu_{Y^2}) & \mu_Y g_i(1-g_i) & g_i(-\mu_Y \mu_{Y^2} + \mu_{Y^3}) & g_i(-\mu_Y^2 g_i + \mu_{Y^2}) \end{bmatrix},$$

where we denote $\mu_{Y^k}$ as the $k$-th moment of $Y$.

Then, using Eq. (17), we have that $1/T \sum_{i=1}^{T} \text{Cov}(w_i) = \sum_{i=1}^{T} \text{Cov}(X_{n,i})$ converges towards $V \in \mathbb{R}^{5 \times 5}$, defined as

$$
V = \begin{bmatrix}
g^{(1)} - g^{(2)} & 0 & g^{(1)} - g^{(2)} & 0 & \mu_Y(g^{(1)} - g^{(2)}) \\
0 & -\mu_Y^2 + \mu_{Y^2} & 0 & -\mu_Y \mu_{Y^2} + \mu_{Y^3} & g^{(1)}(-\mu_Y^2 + \mu_{Y^2}) \\
g^{(1)} - g^{(2)} & 0 & g^{(1)} - g^{(2)} & 0 & \mu_Y(g^{(1)} - g^{(2)}) \\
0 & -\mu_Y \mu_{Y^2} + \mu_{Y^3} & 0 & -(\mu_{Y^2})^2 + \mu_{Y^4} & g^{(1)}(-\mu_Y \mu_{Y^2} + \mu_{Y^3}) \\
\mu_Y(g^{(1)} - g^{(2)}) & g^{(1)}(-\mu_Y^2 + \mu_{Y^2}) & \mu_Y(g^{(1)} - g^{(2)}) & g^{(1)}(-\mu_Y \mu_{Y^2} + \mu_{Y^3}) & -\mu_Y^2 g^{(2)} + \mu_{Y^2} g^{(1)}
\end{bmatrix}.
$$

We have completed the first step of the proof.

Now we want to show that $X_{n,i}$ satisfies the Lindeberg condition (Eq. (12)). Let $\varepsilon > 0$. Because $X_i, Y_i \in [0,1]$, we have that for all $i \leq n$, $\|X_{n,i}\| \leq \sqrt{\frac{10}{n}}$. There exists $n_0 > 0$ such that $\forall n \geq n_0, \sqrt{\frac{10}{n}} < \varepsilon$. Therefore, $\forall n \geq n_0, \forall k \leq n, \mathbb{1}\{\|X_{n,k}\| > \varepsilon\} = 0$. So, for all $n \geq n_0$,

$$
\sum_{k=1}^{n} \mathbb{E}[\|X_{n,k}\|^2 \mathbb{1}\{\|X_{n,k}\| > \varepsilon\}] = 0. \tag{20}
$$

Hence, we have shown that for all $\varepsilon > 0$,

$$
\sum_{k=1}^{n} \mathbb{E}[\|X_{n,k}\|^2 \mathbb{1}\{\|X_{n,k}\| > \varepsilon\}] \to 0. \tag{21}
$$

Therefore, using the Lindeberg CLT (Theorem H.1), we have that

$$
\frac{1}{\sqrt{T}} \sum_{i=1}^{T} (w_i - \mathbb{E}(w_i)) \xrightarrow{d} \mathcal{N}(0, V). \tag{22}
$$

We have completed the second step of the proof. Now, we want to apply the Delta method (Theorem H.2) to show that $S(\omega)$ is normally distributed.

Let $\mu_w := \lim_{T \to \infty} 1/T \sum_{i=1}^{T} \mathbb{E}[w_i] = (g, \mu_Y, g, \mu_{Y^2}, g\mu_Y)$. We introduce

$$
E_i = \frac{1}{\sqrt{T}} \sum_{i=1}^{T} \mathbb{E}[w_i] - \mu_w \tag{23}
$$

$$
= \sqrt{T}\left(\frac{1}{T} \sum_{i=1}^{T} \mathbb{E}[w_i] - \mu_w\right) \tag{24}
$$

$$
= O\left(\frac{1}{\sqrt{T}}\right) \text{ (Using Eq. (17)).} \tag{25}
$$

Therefore, we have

$$
\frac{1}{\sqrt{T}} \sum_{i=1}^{T} (w_i - \mu_w) = \frac{1}{\sqrt{T}} \sum_{i=1}^{T} (w_i - \mathbb{E}[w_i]) + E_i \xrightarrow{d} \mathcal{N}(0, V). \tag{26}
$$

Let $u : \mathbb{R}^5 \to \mathbb{R}$ be defined as

$$
u(x) = \frac{x_5 - x_1 x_2}{\sqrt{(x_3 - x_1^2)(x_4 - x_2^2)}}. \tag{27}
$$

We have that $S(\Omega) = u\left(1/T \sum_{i=1}^{T} w_i\right)$ (using Eq. (19)) and $u(\mu_w) = 0$, and therefore using the Delta method (Theorem H.2) we have that

$$
\sqrt{T} S(\Omega) \xrightarrow{d} \mathcal{N}\left(0, \nabla u(\mu_w)^T V \nabla u(\mu_w)\right). \tag{28}
$$

Because $\nabla u(\mu_w)^T V \nabla u(\mu_w) = 1$, we have shown that

$$
\sqrt{T} S(\Omega) \xrightarrow{d} \mathcal{N}(0, 1). \tag{29}
$$

$\square$

## I  EXTENDED DISCUSSION OF THE STATE OF WATERMARK SPOOFING

In this section, we provide an overview of the state of the field of watermark spoofing to further motivate our work and highlight its practical implications. In App. I.1, we identify three categories of spoofing techniques and highlight learning-based spoofing techniques as the most practically relevant. We put our findings in this context, discussing the potential for adaptive spoofing that does not leave artifacts in the spoofed text. In App. I.2 we discuss how latest schemes attempt to tackle the issue of spoofing.

### I.1  APPROACHES TO SPOOFING

**Learning-based spoofing**    As explained in §1, *learning-based spoofing* operates in two phases. In the first phase, the spoofer queries the model to generate a dataset $\mathcal{D}$ of $\xi$-watermarked text. From this dataset $\mathcal{D}$, the spoofer learns the watermark, which allows them to generate spoofed text. In the second phase, using their knowledge and a private LM, the spoofer can generate *arbitrary* watermarked text *at scale*, without having to query the original model again. In particular, spoofed texts can be created as answer to any prompt, even the one that would be refused by the original LLM, which gives learning-based spoofers great flexibility, and illustrates the potential threat they pose. Additionally, as long as the cost of the first phase is reasonable, learning-based spoofing is cost-effective, as the subsequent per-spoofed-text cost is zero. Learning-based spoofing includes the works of Jovanović et al. (2024); Gu et al. (2024); Zhang et al. (2024).

**Piggyback spoofing**    A second family of spoofing techniques is *piggyback spoofing*, introduced by (Pang et al., 2024), which directly exploits the desirable robustness property of the watermarks. Given a $\xi$-watermarked sentence, the attacker modifies a few tokens to alter the meaning of the original sentence while maintaining the watermark, interpreting the result as an instance of spoofing. While illustrating the potential drawbacks of high robustness, this comes with several caveats. First, abusing the robustness of the watermark naturally raises the question of the boundary between spoofed text and edited $\xi$-watermarked text. Indeed, mixing human and LM text is a realistic use of LMs, and it is agreed that watermarks should account for this use (Kirchenbauer et al., 2023; Kuditipudi et al., 2023). Second, piggyback spoofing is limited in the scope of text it can generate, as it relies on the original model to generate the majority of the text. This greatly reduces the flexibility of the attack, i.e., does not allow the attacker to generate texts on harmful topics that would be refused by the watermarked model. Finally, the same property makes the cost of spoofing scale with the number of spoofed texts, as the attacker needs to query the original model each time.

**Step-by-step spoofing**    A separate line of works considers spoofing techniques that require queries *at each step* of the generation process of *every spoofed text* (Pang et al., 2024; Zhou et al., 2024; Wu & Chandrasekaran, 2024), using the feedback obtained this way to choose the next token. While they have higher flexibility compared to piggyback spoofing, a key limitation of these techniques is the high cost, even compared to piggyback spoofing. Further, some of these methods assume access to the watermark detector itself (sometimes also its confidence score) to obtain the desired feedback, which is not always realistic. For instance, in the case of the first public large-scale deployment of a watermark, SynthID-Text, Google does not provide public access to the watermark detector.

**Summary and our impact**    In summary, learning-based spoofing is the most practically relevant category of spoofing techniques, as it is cost-effective, flexible, and does not require querying the original model for each spoofed text. Another advantage from the perspective of our research question is the fact that current learning-based methods are based on fundamentally different principles, making the question of their common limitations relevant and interesting. In this work, we study that question, showing that all learning-based spoofers leave visible artifacts in spoofed text, which can be leveraged to distinguish between spoofed and $\xi$-watermarked text.

Up to now, the established paradigm in watermark spoofing literature has been to measure the success rate as the percentage of generated texts that succeed in fooling the watermark detector, potentially with certain quality constraints. Our insights imply that producing high-quality spoofed text is a bigger challenge, and motivate future research on avoiding observable artifacts. One promising direction can be work on adaptive spoofers, that build the dataset $\mathcal{D}$ in a way that minimizes the amount of artifacts in spoofed text. We leave this direction open for future work.

## I.2 SPOOFING-AWARE WATERMARKING SCHEMES

The field of watermarking is evolving rapidly, as explained in §6, with different schemes proposed in the literature. We distinguish two approaches to watermarks in LMs. The first one is the statistical approach, notably including schemes from Kirchenbauer et al. (2023); Kuditipudi et al. (2023); Aaronson (2023), which place great emphasis on watermark robustness and practicality. The second is the cryptographic approach, with schemes stemming from Christ et al. (2024), which focus primarily on watermark security and rigorous guarantees.

In particular, schemes with cryptographic features have not been shown to be vulnerable to spoofing attacks. Yet, they enhance security by trading off other key watermark properties, such as robustness to watermark removal. Moreover, recent work (Zhou et al., 2024) suggests merging both fields to create a watermarking scheme that is not only more robust to watermark removal but also to watermark spoofing. However, they show that their approach trades off with generation quality. This highlights that, from the perspective of a model provider, there is no single scheme that is the most desirable. Hence, choosing a particular scheme is a complex task that involves navigating the tradeoffs between different properties. From this perspective, our work provides new evidence that schemes derived from (Kirchenbauer et al., 2023) are harder to spoof than previously thought (as these attempts can be detected by observing the artifacts), and can help model providers adjust their expectations.

