# OpenReview forum: "Discovering Clues of Spoofed LM Watermarks"
_ICLR.cc/2025/Conference — Submitted to ICLR 2025_

### Official Review · Reviewer_n7sN · 2024-10-29

**Soundness:** 3
**Presentation:** 2
**Contribution:** 2
**Rating:** 6
**Confidence:** 3

**Summary:**

This paper proposes a hypothesis-test-based method to detect whether a watermarked text generated by an LLM is spoofed.

**Strengths:**

1. The proposed method can detect spoofed texts when the length of the text is large enough.
2. Detailed theoretical analysis.

**Weaknesses:**

1. The insight of this paper depends on that 'if the prior contexts are not in the adversary's extracted dataset, the adversary will predict the next token randomly'. If so, utilizing a long context as the seed to determine the green-red split may easily defend the spoofing attack. The original watermark verification method (used in the watermarking scheme) may also be able to distinguish the true watermarked texts from the spoofed watermark texts. Further clarifying the challenges of detecting spoofed texts may help the readers better understand the technical contribution of this paper.
2. Another concern is that this paper does not include any baseline method in the experiments. I think comparing with some baseline methods (e.g., the original watermark verification method) can help demonstrate the effectiveness of the proposed method.
3. Section 5.1 presents the histograms of the distributions. However, a quantitative analysis (e.g., a hypothesis test to verify whether the distribution follows a normal distribution) can better help the readers understand the results.
4. This paper may not be easy to read for readers who are not familiar with the statistics. I strongly suggest that the authors explain some uncommon symbols and operators. For example, the symbol in Eqs. (3a) and (3b) and the $\xrightarrow{d}$ in Lemma 4.1.

**Questions:**

Please refer to the weakness section.

---

> ### Author Response · Authors · 2024-11-22
> **Response to Reviewer n7sN**
>
> We thank the reviewer for their exhaustive feedback. Below, we address the concerns raised in questions Q1 to Q4; and note that a revised version of the paper has been uploaded, with updates highlighted in blue. We are happy to clarify further if there are additional questions.
>
> **Q1: Could the authors clarify why simply increasing the context size $h$ isn’t enough to overcome spoofing?**\
> Certainly. In fact, several previous works ([1,2]) have already noted that using a longer context size $h$ does make spoofing harder. In fact, the Distillation spoofing technique [3] does not work with $h=3$. However, previous works ([4,5,6]) have already noted that the choice of $h$ is a complex tradeoff between many watermark properties such as watermark robustness, impact on quality, or ease of spoofing. In general, higher $h$ values are harder to spoof  and have less impact on the text quality at the cost of being more vulnerable to scrubbing attacks.
>
> We agree with the reviewer that such a discussion regarding $h$ is missing in our paper. We edited the introduction to include this discussion. We are happy to further edit the paper should the reviewer have other suggestions.
>
> **Q2: Why don’t the authors include the original watermark detector as the baseline?**\
> We believe this is a misunderstanding. As explained in Section 3, our threat model is that the texts on which our statistical test is applied are already flagged as watermarked by the original watermark detector. This is a realistic scenario: a model provider does not care about investigating texts if they don’t pass the detector, as such a text is unrelated to their model (e.g., human written or a failed spoofing attempt). We hope this clarifies a potential misunderstanding. Similarly, there are no prior “spoofing discovery” baselines, as we are the first to study the detectability of spoofing attempts. We are happy to clarify our answer if it does not fit the reviewer's question.
>
> **Q3: Could the authors provide a quantitative analysis of the validation of the test hypotheses?**\
> In our new revision, we updated Figure 2 in Section 5.1 to add quantitative information regarding the histograms normality. More specifically, we ran both a Kolmogorov-Smirnov test for standard normality and a Pearson’s normality test. For the Standard method, the tests confirm the visual intuition that only $h=3$ follows a standard normal distribution. For the Reprompting method, we observe that for $h \ge 2$, the standard normal hypothesis is only slightly violated (p-values of the KS test are greater than 0.001), and for $h=1$, only the normality hypothesis holds. The slight violations of the hypothesis for the Reprompting method are expected, as it does not have a theoretical backing as strong as the Standard method. Yet, in practice, the FPR of the tests is properly controlled for both methods, as we show in Section 5.2.
>
> We thank the reviewer for their suggestion, and we believe the additional quantitative analysis coupled with the histograms strengthens our message that our hypotheses are sound and practical.
>
> **Q4: Could the authors clarify the symbols used in the paper?**\
> Good point, we updated the paper to ensure all less common symbols are now properly introduced.
>
> The symbol in equation (3a) and (3b) means independence and the symbol $\xrightarrow{d}$ is used for convergence in distribution.
>
> We are happy to further edit the paper if the reviewer has more suggestions.
>
> [1] “On the Reliability of Watermarks for Large Language Models”, Kirchenbauer et al., ICLR 2024\
> [2] “Watermark stealing in large language models”, Jovanovic et al., ICML 2024\
> [3] “On the learnability of watermarks for language models”, Gu et al., ICLR 2024\
> [4] “A watermark for large language models”, Kirchenbauer et al., ICML 2023\
> [5] “Provable robust watermarking for ai-generated text”, Zhao et al., ICLR 2024\
> [6] “Robust distortion-free watermarks for language models”, Kuditipudi et al., TMLR 05/2024

---

> ### Comment · Reviewer_n7sN · 2024-11-25
>
> Thank you for the detailed response. It addresses most of my concerns. I decide to raise my rating to 6. I do not give a higher score due to the following concerns.
>
> 1. This paper only investigates one watermarking method (i.e., red-green watermarks).
> 2. The contributions of this paper are to provide insight to discover spoofed watermarks and design a hypothesis test. However, this paper does not explicitly clarify the challenges of designing such a method. Therefore, I think this paper may provide limited inspiration for subsequent works. I think designing a trivial method as a baseline may be better to clarify the challenges and the advantages of the designed method. In summary, I think this paper can provide some insight to the community but the technical contribution of this paper is limited.

---

> > ### Author Response · Authors · 2024-11-25
> >
> > We thank the reviewer for raising their score to recommend acceptance. Regarding the newly raised concern (focus on red-green watermarks), we kindly refer the reviewer to our answer to Q1 of Rev. KEcn where we motivate our focus, and our new Appendix F, where we demonstrate that spoofing of AAR and KTH schemes also leaves artifacts.

---

### Official Review · Reviewer_KEcn · 2024-10-31

**Soundness:** 3
**Presentation:** 3
**Contribution:** 3
**Rating:** 5
**Confidence:** 4

**Summary:**

This paper presents the first in-depth study of statistical artifacts in LLM watermark spoofing attacks. The authors demonstrate that current watermark spoofing methods leave detectable statistical patterns and propose a statistical testing framework to distinguish genuine watermarked text from spoofed ones.

**Strengths:**

1. First systematic investigation of quality characteristics in watermark spoofing, beyond mere success rates
2. Strong theoretical foundation with rigorous statistical testing framework
3.  Practical approach for detecting watermark spoofing, enhancing watermark reliability

**Weaknesses:**

1. The study's scope is primarily confined to the KGW watermark algorithm, leaving significant gaps in the evaluation of other important watermarking approaches. The effectiveness of the proposed method remains untested on unigram watermarking schemes  (zhao et al. 2023) that employ global green/red lists, as well as the KTH  (Kuditipudi et al. 2023)  method utilizing fixed watermark key lists. This limited coverage raises questions about the method's generalizability to alternative watermarking techniques widely used in practice.
2.  The proposed approach imposes dual requirements of maintaining both significant green/red word ratios and cross-context consistency, which may compromise its robustness. While these requirements enhance the detection accuracy of spoofed watermark text, they could make the system overly sensitive to legitimate text modifications. Because text modification will also cause not consistent context watermark. The paper does not adequately address the critical trade-off between detection accuracy and robustness against common text editing operations. Further investigation is needed to determine whether the method can maintain its effectiveness while accommodating natural text modifications.
3. The method's requirement for approximately 1000 tokens to achieve reliable detection presents significant practical limitations. This substantial text length requirement restricts the method's applicability in real-world scenarios, particularly for short-form content analysis.

**Questions:**

Please refer to the weakness part.

---

> ### Author Response · Authors · 2024-11-22
> **Response to Reviewer KEcn**
>
> We thank the reviewer for their exhaustive feedback. Below, we address the concerns raised in questions Q1 to Q3; and note that a revised version of the paper has been uploaded, with updates highlighted in blue. We are happy to clarify further if there are additional questions.
>
> **Q1: Why do the authors restrict themselves to Red-Green watermarks? Does this generalize to other spoofed schemes?**\
> We do not see our focus on Red-Green schemes as a weakness of our work. To be able to evaluate our test, we can only work with schemes where practical spoofing techniques have already been demonstrated. Hence, the only schemes we can study are Red-Green schemes [1], AAR [2] and KTH [3].
>
> In our main experiment, we focused on Red-Green schemes [1] as they are not only the most popular [6] class of watermarks, but are also seeing increasing adoption by industry (e.g., SynthID recently being announced for Gemini). As the most studied class of schemes, Red-Green also happens to have the most research on spoofing attacks, being the only one with more than one learning-based spoofing attack being proposed. As discussed in Section 7, we excluded the Unigram scheme [4] as the spoofing method from [5] achieves perfect spoofing on it (they recover up to 80% of the red-green split).
>
> To highlight the method's generalizability to alternative watermarking techniques, we provide in Appendix F two additional experiments to encompass AAR [2] and KTH [3] with Distillation based spoofing. Importantly, we have similar results as in Section 5, **showing that for both schemes, Distillation leaves artifacts in the generated text** that we can leverage to detect the spoofing attempts.
>
> We hope the additional experiments convince the reviewers that the core idea behind our methods can be generalized to a wider array of schemes.
>
> **Q2: What is the performance of the method under common text modifications?**\
> Our understanding of the reviewer's comment is that they are unsure how the FPR scales when $\xi$-watermarked text is modified. For spoofed text, a potential spoofer would not want to modify their text in a way that lowers the watermark strength, as they precisely aim to make it as strongly watermarked as possible.
>
> We however added a new experiment in Appendix G where we inserted a fixed percentage of human text into a $\xi$-watermarked text. We show that up to 20% of mixed human text within a $\xi$-watermarked text does not significantly affect the FPR of our detection tests for both the Reprompting and Standard methods.
>
> Such results have intuitive explanations: because the alterations to the watermarked text are done independently of the color, the overall correlation within the edited text between the color and the general token frequency remains unchanged. In other words, the tests do not focus on the green/red ratio but on the consistency of green tokens: in a spoofed text, the green tokens are mostly “generally frequent” tokens, whereas in other cases any token is equally likely to be green.
>
> We hope that this additional experiment convinces the reviewer that the tests are robust to legitimate text modifications. We are happy to provide further testing if the reviewer has other suggestions.
>
> **Q3: Does the requirement of 1000 tokens pose a practical limitation?**\
> No. First, as we have shown in Appendix B, the test results remain valid if we concatenate multiple texts of different lengths. Hence, in a practical scenario, a longer text could be directly achieved by concatenating multiple texts from the same source. More generally, we believe the key contribution of our work is already highlighting the existence of artifacts in current learning-based spoofing methods.
>
> [1] “A watermark for large language models”, Kirchenbauer et al., ICML 2023\
> [2] “Watermarking of large language models”, Scott Aaronson, 2023 Workshop on Large Language Models and Transformers, Simons Institute, UC Berkeley\
> [3] “Robust distortion-free watermarks for language models”, Kuditipudi et al., TMLR 05/2024\
> [4] “Provable robust watermarking for ai-generated text”, Zhao et al., ICLR 2024\
> [5] “Large Language Model Watermark Stealing With Mixed Integer Programming”, Zhang et al., 2024\
> [6] “A survey of text watermarking in the era of large language models”, Liu et al., ACM Computing Surveys 2024

---

> > ### Comment · Reviewer_KEcn · 2024-11-25
> >
> > I find the response to Q1 generally satisfactory. However, I believe the response to Q2 does not address my original concern. While the paper uses consistency of green tokens for detecting spoofed text, which is reasonable, another important property of watermarking is robustness - the ability to detect watermarks even after text modifications. The question remains whether modified text might also show inconsistency in green tokens. The current evaluation only considers copy-paste attacks, which is insufficient. The authors should further examine word deletion, substitution, and even paraphrasing to determine if legitimate watermarked text (not spoofed) after modification can be distinguished from spoofed watermarked text. If such distinction cannot be made, this method may be essentially identifying both spoofed text and modified watermarked text together, making it impossible to make a 100% determination. While I acknowledge that some sacrifice in robustness is inevitable, the paper should include an appropriate discussion of these limitations.
> >
> > The response to Q3 is also not entirely satisfactory. The main issue lies in what exactly is the threat model in the current problem. Do users only have access to the text being detected, or can they also access the target LLM? If the problem is to detect watermarks given only a text sample, we typically cannot make multiple requests to an LLM to obtain multiple text samples from the same source. The threat model in this detection process needs to be more clearly defined.

---

> > > ### Author Response · Authors · 2024-11-26
> > >
> > > We thank the reviewer for engaging in the discussion.
> > >
> > > **Q2.** We agree with the reviewer that distinguishing between modified $\xi$-watermarked text and spoofed text is an important consideration.
> > > We argue that the case of deletion/substitution is already included in the copy-paste experiment, as these can be seen as fine-grained variants of copy-pasting. Moreover, in prior works ([1,2]), the experimental process to study word-level deletion/substitution is to naively delete/substitute tokens at random, without any consideration for the meaning of the text. Hence, such an experimental process is not particularly relevant in our case as fluency matters: it is hard to find a practical use-case where a model provider would like to know the origin of nonsensical text.
> > >
> > > For this reason, many prior works [1, 2] also evaluate robustness to paraphrasing, which the reviewer also mentions. It is a strictly harder and more meaningful scenario, as it implies more changes while resulting in natural text. Following the experimental setting from [1], we now extended our Appendix G with an experiment that evaluates the FPR of paraphrased $\xi$-watermarked text using state-of-the-art paraphraser Dipper-11B. We obtained similar results to the copy-paste experiment: the Type 1 error of the test remains controlled, i.e., paraphrased $\xi$-watermarked text is not falsely flagged as spoofed. We believe this makes sense intuitively as the paraphrasing is done independently of the color/frequency of the tokens and should not have abnormal correlation.
> > > We are happy to provide further testing if the reviewer has other suggestions.
> > >
> > > **Q3.** We want to clarify the threat model, and we have updated Sections 3 and 4 in our paper accordingly. We assume we have access to a set of $K$ texts from the same source that are labeled as watermarked by the original watermark detector. To detect if the source is spoofing our watermark, we concatenate the $K$ texts and run our test. In the special case of $K=1$, if the single text is short, our test might fail to distinguish whether the watermark is spoofed. For instance, at short text length of 500 tokens, the best parameters show a TPR of 62% at a confidence level of 99%. This shows some robustness to short texts, but reducing text length further may be a challenge—in our latest revision, we added this to the limitations to highlight it as an important consideration.
> > >
> > >
> > > [1] “Watermarking of large language models”, Scott Aaronson, 2023 Workshop on Large Language Models and Transformers, Simons Institute, UC Berkeley\
> > > [2] “Robust distortion-free watermarks for language models”, Kuditipudi et al., TMLR 05/2024

---

> > > > ### Author Response · Authors · 2024-12-01
> > > >
> > > > We thank the reviewer again for following up. Did our new paraphrasing experiment and threat model clarifications help resolve the remaining concerns? If so, we would appreciate if the reviewer could consider raising their score.

---

### Official Review · Reviewer_yZgn · 2024-11-01

**Soundness:** 3
**Presentation:** 3
**Contribution:** 2
**Rating:** 6
**Confidence:** 4

**Summary:**

This work proposes statistical tests that can differentiate spoofed texts from genuine watermarked texts, based on the intuition that spoofed text contains artifacts due to spoofers' limited knowledge.

**Strengths:**

1. This work investigates an interesting problem of whether the spoofed texts can be differentiated even if they trick the detector.

2. It provides a theoretical demonstration regarding the proposed tests.

3. The experiments are extensive, testing different language models and diverse settings.

**Weaknesses:**

1. The conclusion is overclaimed. This paper only examined learning-based spoofing attacks but claims that all state-of-the-art spoofing attacks leave artifacts in spoofed texts (in the abstract and introduction). What about spoofing attacks that exploit watermark robustness, e.g., Peng et al. [1] and Zhou et al. [2] in your references?

2. The proposed tests are not practically useful, considering the performance is only good in some cases. In particular, from Table 1, the tests yield high TPR when $ T > 1000 $ and $ h = 3 $ or $ h = 1 $. However, spoofed texts from two learning-based attacks could achieve spoofing attacks when $ T < 1000 $. If their generated text has $ T = 500 $, the corresponding test performance at best is 0.62 (TPR@1%). Does this suggest that the model owner should deploy watermarking with $ h = 3 $?

3. There is no explanation of the important variables or unclear definitions, impeding understanding (see question 1).


4. Limited discussions. Intuition suggests that the reason for artifacts is that the spoofer only has partial knowledge due to their limited training data. How will the size of $ D $ affect the artifacts and the statistical test results? Would stronger attackers leave fewer artifacts by increase the size of $D$ or improving the sampling strategy by introducing some independence?






----
[1] Qi Pang, Shengyuan Hu, Wenting Zheng, and Virginia Smith. Attacking LLM watermarks by exploiting their strengths. arXiv, 2024.

[2] Tong Zhou, Xuandong Zhao, Xiaolin Xu, and Shaolei Ren. Bileve: Securing text provenance in large language models against spoofing with bi-level signature. arXiv, 2024.

**Questions:**

### Questions in Weaknesses and Below:


1. What is $X_t$ (is it the same as defined in line 168?), and what is the meaning of $X_t = 1 $ in equations 2a and 2b? Also, why is $I_D $ defined as a function of frequencies in line 194 but as a probability distribution in line 204? Which is the correct definition?

2. How accurate is it to use $\tilde{D} $ to approximate the training set $D $ (i.e., using natural language to approximate watermarked language)? To what extent could the discrepancy affect the final accuracy of distinguishing spoofed text?

3. How many samples are in the spoofer training data $D $? There are two different distillation methods proposed in Gu et al. (2024); which one did you test?

---

> ### Author Response · Authors · 2024-11-22
> **Response to Reviewer yZgn**
>
> We thank the reviewer for their exhaustive feedback. Below, we address the concerns raised in questions Q1 to Q9; and note that a revised version of the paper has been uploaded, with updates highlighted in blue. We are happy to clarify further if there are additional questions.
>
> **Q1: Do the authors claim any spoofing attempts leave artifacts? What about non-learning-based spoofing techniques?**\
> It is not our intention to make such overclaims and are happy to adjust the claims in order to improve clarity in this regard. In particular, we do not claim that every spoofing method leaves artifacts in their generated text. Instead we argue that current practical state-of-the-art spoofers for popular watermarking schemes both (1) happen to be learning-based in nature and (2) do in fact leave artifacts of watermark spoofing in their generated texts.
>
> Notably we focus on learning-based spoofing as they have the ability to generate arbitrary amounts of diverse spoofed text in a cost-effective way and are applicable in realistic setups. We see this in contrast to the methods mentioned by the reviewer. In particular, “Piggyback spoofing attacks” ([1]) directly leverage the desirable robustness property of the watermark. Given a watermarked sentence, the attacker modifies a few tokens to alter the meaning of the original sentence while maintaining the watermark, interpreting the result as an instance of spoofing. Mixing human and LM generated text is a realistic use case of LMs, many watermarking methods inherently account for such modifications ([2,3]), making it a priori unclear whether selectively modifying individual tokens is practical spoofing or an inherent property of a watermark. We note that this is also in stark contrast to learning-based spoofers that can generate almost arbitrary text without having to rely on a pre-existing template—which additionally can be both hard to generate for harmful topics and significantly increases costs as it requires queries to the original LM for every generation.
>
> A separate line of works considers spoofing techniques that require queries at each step of the generation process of every spoofed text [1,4,5], using the feedback obtained this way to choose the next token. While they have higher flexibility compared to piggyback spoofing, a key limitation of these techniques is the prohibitively high cost, even compared to piggyback spoofing. Further, some of these methods rely on the unrealistic assumption of having access to the watermark detector itself (sometimes even its confidence score) to obtain the required feedback, making them inapplicable to practical deployment scenarios (such as the recently released SynthID-Text).
>
> Hence, our focus on learning-based spoofing is largely driven by practical considerations, as well as the state of the field regarding spoofing attacks. To clarify our contribution, we have added this additional discussion to the new Appendix I and are happy to adjust claims to accurately reflect the contributions of our work.
>
>
> **Q2: Does the requirement of 1000 tokens pose a practical limitation?**\
> No. First, as we have shown in Appendix B, the test results remain valid if we concatenate multiple texts of different lengths. Hence, in a practical scenario, a longer text could be directly achieved by concatenating multiple texts from the same source. More generally, we believe the key contribution of our work is already highlighting the existence of artifacts in current learning-based spoofing methods.
>
> **Q3: Can the test results be used to choose a context size?**\
> Partially, but we do not make confident claims about which $h$ is best. Our results, that current learning based spoofing attacks leave visible artifacts in spoofed text, show that the risk of spoofing for smaller $h$, with current methods, is less important than previously thought. However, previous works ([2,6,7]) have already noted that the choice of $h$ is a complex tradeoff between many watermark properties such as watermark robustness, impact on quality, or ease of spoofing. In general, higher $h$ values are harder to spoof  and have less impact on the text quality at the cost of being more vulnerable to scrubbing attacks. As such we don’t deem it recommendable to make the choice of $h$ solely based on our tests.

---

> ### Author Response · Authors · 2024-11-22
>
> **Q4: Could increasing the size of the spoofer training dataset significantly degrade the efficiency of the method?**\
> We agree with the reviewer that the intuition behind our test is that the artifacts left by  spoofing are caused by the spoofer's partial knowledge of the scheme. Hence, intuitively, a larger training dataset should imply fewer artifacts.
>
> To further investigate this, we ran additional experiments in Appendix E with Stealing on LeftHash with $h=3$. Surprisingly, we find that, when ranging from 4 million to 24 million tokens in $\mathcal{D}$, the TPR of the test is similar. At the same time, the number of successful spoofing attempts increases significantly up to 16 million tokens and then plateaus. This suggests that either the additional data in $\mathcal{D}$ is redundant or that Stealing doesn’t leverage the dataset efficiently. Hence, we believe that preventing the artifacts from Stealing would require an unreasonably large dataset $\mathcal{D}$.
>
> **Q5: Could a more elaborate spoofing technique build the training dataset $\mathcal{D}$ adaptively?**\
> We believe that querying the $\xi$-watermarked model adaptively to develop a spoofing technique that leaves no artifacts is an interesting direction that may be possible, as we suggested in Section 7. However, a significant amount of work is needed to actually design such a technique and evaluate its efficiency. We are excited that our work opens a new challenge in the field of watermark spoofing and hope it can serve as a baseline for evaluating future spoofing techniques.
>
> **Q6: Could the authors provide experiments on the impact of $\tilde{\mathcal{D}}$?**\
> Good question—we added two detailed experiments in Appendix D. For the first experiment, we build several $\tilde{\mathcal{D}}$ by adding random noise to $\mathcal{D}$. This allows us to precisely control the distance between $\tilde{\mathcal{D}}$ and $\mathcal{D}$; and see how the power decreases with the distance. For the second experiment, we show that for realistic choices of $\tilde{\mathcal{D}}$ (e.g., C4, Wikipedia), the power of the test remains very similar to the best case scenario of $\tilde{\mathcal{D}} = \mathcal{D}$. This means that ultimately the choice of $\tilde{\mathcal{D}}$ is not of great importance to maintain high power.
>
> **Q7: Could the authors clarify some points in the paper?**\
> We have edited the paper to clarify the highlighted points, and we hope it is now clearer. We welcome additional suggestions from the reviewer to improve the writing quality of the paper. Below is a short description of the changes.
>
> In our work, $X_t$ (defined in line 168) is the random variable associated with the color of the $t$-th token in $\Omega$. Hence, $X_t = 1$ is the event “the $t$-th token is green”. It is the same $X_t$ used for equations (2a) and (2b).
>
> We implicitly consider that $I_{\mathcal{D}}$, as a function of frequencies, defines a categorical probability distribution over its support, where the probability of an $h+1$-gram is equal to its frequency. Based on the reviewer's suggestion, we removed this distinction from our paper, as it did not add value to the comprehension of the proposed method.
>
> **Q8: How many samples are in $\mathcal{D}$?**\
> We follow the recommendations from the original papers. For Stealing ([8]), we used 30,000 prompts, each 800 tokens long, based on completions of C4. For Distillation ([9]), we used 640,000 prompts, each 256 tokens long, based on completions of C4. Based on the reviewer’s comment, we clarified these numbers in the paper (in Section 5, L368).
>
> **Q9: Which method did you use for Distillation spoofing?**\
> In [9], the two methods presented are distillation from a watermarked teacher model or from samples generated by the watermarked model. In a spoofing case, as the authors of [9] explain, we assume we do not have access to the original model which is required for the first scenario. Hence, we focus only on the second method based on learning from a sampled dataset of $\xi$-watermarked text. We updated the paper (Section 2) to specify which method we used and avoid any future confusion.
>
>
> [1] “Attacking LLM Watermarks by Exploiting Their Strengths”, Pang et al., 2024\
> [2] “A watermark for large language models”, Kirchenbauer et al., ICML 2023\
> [3] “Robust distortion-free watermarks for language models”, Kuditipudi et al., TMLR 05/202
> [4] “Bileve: Securing Text Provenance in Large Language Models Against Spoofing with Bi-level Signature”, Zhou et al., 2024\
> [5] “Bypassing LLM Watermarks with Color-Aware Substitutions”, Wu et al., 2024\
> [6] “Robust distortion-free watermarks for language models”, Kuditipudi et al., TMLR 05/2024\
> [7] “Provable robust watermarking for ai-generated text”, Zhao et al., ICLR 2024\
> [8] “Watermark stealing in large language models”, Jovanovic et al., ICML 2024\
> [9] “On the learnability of watermarks for language models”, Gu et al., ICLR 2024

---

> > ### Comment · Reviewer_yZgn · 2024-11-22
> >
> > Thank you to the authors for the prompt revisions and for providing additional experiments, which address most of my concerns. However, I still have a few remaining concerns outlined below:
> >
> > ---
> >
> >  **1. Clarify the Scope in the Abstract and Introduction**
> >
> > The scope of your work should be more explicitly defined. Specifically, you focus on **learning-based spoofing attacks**, but the abstract and introduction use vague terms like `current state-of-the-art spoofing,` which could lead to confusion. If you intend to make such broad claims, please provide additional results on the other two types of spoofing attacks summarized in Appendix I. Otherwise, revise these sections to clearly reflect the scope and update the manuscript accordingly.
> >
> >
> >
> > **2. Handling Short Text**
> >
> > If the text length is only a few hundred tokens, does your method reliably detect spoofed text if the short text is repeated multiple times and concatenated? Alternatively, does your method have limitations in detecting spoofed text when the input is a single short text? Please clarify this limitation and its implications.
> >
> >
> >
> > **3. Concerns Regarding Q4 and Figure 9**
> > #### **3.1. Calculation of  D**
> > How is the size of D calculated? Is it |D|*len_query or |D|*T? Where does 4 million come from? Does your conclusion about the TPR behavior originate from Figure 9? Specifically, I could not verify your rebuttal's claim: *"When ranging from 4 million to 24 million tokens in D, the TPR of the test is similar."*
> >
> > #### **3.2. Analysis of Figure 9**
> > Could you analyze Figure 9 for a fixed $\alpha$ value, such as $10^{-3}$? The results raise additional questions:
> > - Does a higher TPR indicate better spoofing detection (correct me if I am wrong)? If so, why does a spoofer trained on a larger D appear to produce text that is easier to detect? For example:
> >    - When $\alpha = 10^{-3}$  and  T = 1000, the TPR is  $\sim0.75$ for |D| = 30,000, but only $\sim0.45$ for |D| = 5,000.
> >    - The claim that "TPR is similar" is difficult to justify when there is a difference of approximately 0.3.
> > - Intuitively, one might expect a spoofer trained on a larger dataset (with more knowledge) to leave fewer artifacts, making detection harder. Can you explain why your results show the opposite behavior?
> >
> > ---
> >
> > These remaining concerns need clarification and additional analysis to strengthen the paper's claims. Thank you for addressing these points.

---

> ### Author Response · Authors · 2024-11-23
>
> We thank the reviewer for the quick response and answer the follow-up questions below.
>
> **1** As suggested by the reviewer, we updated the Abstract, Introduction and Conclusion and uploaded the new revision. We hope that the updates remove any confusion in this regard.
>
> **2** We first want to clarify that concatenating multiple texts (as studied in Appendix B) is only statistically valid in the case of independent texts. Hence, it is not sound to concatenate repetitions of a single short text.
>
> As we state in Section 5 and show in Table 1, the power of the test indeed increases with the length of the text. Importantly, the test remains sound for short texts (i.e., the Type 1 error is still properly controlled), but the chance of false negatives is higher. However, as also noted by the reviewer, at short text length of 500 tokens, the best parameters show a TPR of 62% at a confidence level of 99%. This shows some robustness to short texts; but reducing text length further may be a challenge—in our latest revision, we added this to the limitations to highlight it as an important consideration.
>
> **3.1** We are sorry for the confusion. As explained in the caption of Figure 9, there we report the number of queries, each query being 800 tokens long. Hence, for the smallest $\mathcal{D}=5,000$ we have $5000\times800=4,000,000$ tokens, and for the largest $\mathcal{D}=30,000$ we have $24,000,000$ tokens.
>
> **3.2** Good observations! We thank the reviewer for looking at our new experiments so thoroughly.
> We believe there are several reasons why it would be misleading to convincingly claim that there is positive correlation between the size of $\mathcal{D}$ and the power of the test:
> - At $T=500$ the differences are negligible; this suggests a complex interplay between $|\mathcal{D}|$ and $T$ that may be hard to fully understand from our experiment.
> - Even for $T=1000$ the distance between curves is not as big for $\alpha \in \\{10^{-2}, 10^{-1}\\}$ as the case reviewer justifiably points out.
> - In the interest of time, this new experiment was run with much less data compared to our original experiments ($n=500$ responses vs $n=10,000$ originally). Moreover, these responses are concatenated in groups to get text length of $T$, and the failed spoofing attempts are filtered out (as we only apply our test on successful spoofs!). Crucially, for $|D|=5000$, as spoofing often fails, this results in only $32$ texts, making our results noisy.
> - Such correlation, if true, would have to hold only up until some point, as it is clear that as $|\mathcal{D}| \to \infty$ our test’s power goes to $0$ (as spoofing is perfect). That point would also heavily depend on the setting, e.g., the spoofer algorithm.
>
> In summary, there are many factors at play that are hard to isolate, including our experimental setting or the way a spoofer utilizes data. Thus, while we are not comfortable making hard statements at this point, it could be true that counterintuitively, reducing $|\mathcal{D}|$ reduces our power.
>
> Connecting to the last point above, our best hypothesis is that for very small $|\mathcal{D}|$ the spoofer may fail to acquire any generalization ability, so the (rare) cases when it succeeds are the cases when it remains very close to its training data, i.e., leaves less artifacts. Ultimately, it is important to conclude that smaller $|\mathcal{D}|$ is not strictly better for the spoofing adversary—while there may be less artifacts, the more important goal of the spoofer (to spoof) is often greatly sacrificed. We updated App. E to reflect the above.

---

> > ### Comment · Reviewer_yZgn · 2024-11-25
> >
> > After reviewing the rebuttal, I have decided to raise my rating to 6. However, a higher score cannot be given due to the following remaining concerns:
> >
> > 1. The test performance on shorter text is limited, which restricts its practical applicability to certain scenarios.
> >
> > 2. The experimental results related to the size of  D  are counterintuitive and require further analysis and clarification.

---

> > > ### Author Response · Authors · 2024-12-01
> > >
> > > We thank the reviewer for raising their score to recommend acceptance. We also appreciate that they explicitly summarize the two aspects in which the paper could be improved, and we will make sure to highlight these more prominently as well.

---

### Official Review · Reviewer_xcg2 · 2024-11-03

**Soundness:** 3
**Presentation:** 3
**Contribution:** 3
**Rating:** 6
**Confidence:** 3

**Summary:**

This paper investigates the vulnerabilities in watermarking LLMs due to spoofing attacks. While watermarks aim to identify AI-generated text, attackers can replicate them, falsely attributing unauthorized content to specific models. By analyzing "artifacts" left by spoofing techniques, the authors develop statistical tests to differentiate between authentic and spoofed watermarked text. Experiments show these tests are effective across various spoofing methods.

**Strengths:**

1. The paper provides a detailed analysis of the specific, observable artifacts left in spoofed texts, an underexplored area in watermark spoofing research.

2. The authors conduct extensive experiments, testing their hypotheses across multiple models, parameter settings, and spoofing techniques, lending empirical robustness to their conclusions.

**Weaknesses:**

1. The paper focuses primarily on a specific watermarking scheme -- the KGW watermark. While this is an important and well-studied case, there are several newer watermarking designs that make spoofing attacks virtually impossible. These alternative methods could be used without the concerns associated with spoofing attacks. As such, the scope of the paper feels limited, as it does not consider the broader landscape of watermarking schemes. Furthermore, the paper mainly studies specific spoofing techniques, such as Stealing and Distillation, which may not fully reflect the evolving and diverse strategies available to attackers.

2.  Some assumptions in the analysis are restrictive, particularly regarding the KGW watermark’s context size, often set at h=1. With this small context, green tokens are more likely to appear in the spoofer's training data, simplifying the spoofing process. For example, the assertion that "If the context is not in D, the spoofer is forced to select the next token independently of its color" mainly applies when h is larger, limiting its relevance to cases with small context sizes.

3. The paper would benefit from a deeper exploration of how the context length h influences the statistical tests’ performance. Additionally, discussing broader applications and limitations for various watermark types could enhance the study’s relevance. For instance, cryptographic watermark designs, such as those based on pseudorandom error-correcting codes, may not face spoofing issues, and a comparison with these designs could provide more context on the strengths and weaknesses of KGW-style watermarks.

**Questions:**

None

---

> ### Author Response · Authors · 2024-11-22
> **Response to Reviewer xcg2**
>
> We thank the reviewer for their exhaustive feedback. Below, we address the concerns raised in questions Q1 to Q3; and note that a revised version of the paper has been uploaded, with updates highlighted in blue. We are happy to clarify further if there are additional questions.
>
> **Q1: Why do the authors restrict themselves to two learning-based spoofing techniques? What about other evolving and diverse strategies?**\
> The scope of our work are learning-based spoofers (as defined in Section 2), as they have the ability to generate arbitrary amounts of diverse spoofed text in a cost-effective way and are applicable in realistic setups. This, given the current state of the field, boils down to two methods: Stealing ([1]) and Learnability ([2]). We excluded the spoofing method from [3], as it achieves perfect spoofing on the Unigram scheme [4] (thus can’t be detected) and does not spoof any other schemes. In Section 6, we had already discussed the works of [5] and [6]. We have further updated our paper (Appendix I) to include an even more thorough discussion of those topics.
>
> We are not aware of any other works beyond the ones we discuss. Could the reviewer point out some specific spoofing techniques they had in mind? We would be happy to discuss them, and see if our methods can generalize.
>
> **Q2:  Is the contribution invalidated by the existence of a non-spoofable scheme? Why do the authors restrict themselves to Red-Green watermarks?**\
> We do not see the existence of harder to detect schemes as a weakness of our work.
> While we agree that cryptographic schemes such as [11] offer strong guarantees against spoofing, watermarking is a complex field which balances many properties such as strength, robustness to removal, or behavior in low entropy settings. Notably many of these properties are not evaluated on such schemes leaving them primarily as theoretical constructs. Based on the reviewer’s suggestion, we expanded this discussion in Appendix I.
>
> In contrast Red-Green schemes [7] are not only the by far most popular [12], but are also seeing increasing adoption by industry (with SynthID recently being announced for Gemini). As the most studied class of schemes, Red-Green also happens to have the most research on spoofing attacks, resulting in several practical attacks [1,2,3]. Combined with the practical adoption of RG schemes, this makes the detection of such spoofed texts a relevant and timely topic of study.
>
> As we elaborate in our new Appendix F, the key ideas behind our spoofing detection test generalize to both AAR [8] and KTH [9], where we show that Distillation of both schemes also leaves detectable artifacts.
>
> We hope that this discussion and the additional results justify our focus on Red-Green watermarks and further highlight how the core ideas behind our method generalize to a wider range of schemes.
>
> **Q3: Could the authors provide additional experiments to show the influence of the context size on the test efficiency? Is spoofing with smaller contexts harder to detect?**\
> We believe this is a misunderstanding as we already evaluated the power of our test for small context sizes.
>
> All our results in Section 5 are presented for $h \in \{1,2,3\}$. For Distillation spoofing, we observe that the test is slightly more powerful for $h=2$ than for $h=1$. However, for Stealing spoofing, counterintuitively, the test is less powerful for $h=2$ than for $h=1$. Hence, our experimental evaluation in Section 5 already shows that a smaller context does not weaken our method. We agree that a bigger context size makes spoofing harder and could prevent spoofing altogether. However, as shown in prior work [7], using larger $h$ to counter spoofing comes with drawbacks: it reduces the robustness of the watermark. In the limit, if $h=\infty$, changing a single token in a text would completely break the watermark, which is not a desirable property. We added this discussion to the updated version of our paper (Section 1).
>
> There is still the special case of $h=0$. This is the Unigram scheme ([4]), which can be almost perfectly (they recover up to 80% of the red-green split) using the method from [3], as we explain in Section 6. Hence, we excluded it from our experiments in the first place.
>
> If the reviewer has specific experiments in mind, we would be happy to include them in our paper.

---

> > ### Author Response · Authors · 2024-11-22
> >
> > [1] “Watermark stealing in large language models”, Jovanovic et al., ICML 2024\
> > [2] “On the learnability of watermarks for language models”, Gu et al., ICLR 2024\
> > [3] “Large Language Model Watermark Stealing With Mixed Integer Programming”, Zhang et al., 2024\
> > [4] “Provable robust watermarking for ai-generated text”, Zhao et al., ICLR 2024\
> > [5] “Bypassing LLM Watermarks with Color-Aware Substitutions”, Wu et al., 2024\
> > [6] “Attacking LLM Watermarks by Exploiting Their Strengths”, Pang et al., 2024\
> > [7] “A watermark for large language models”, Kirchenbauer et al., ICML 2023\
> > [8] “Watermarking of large language models”, Scott Aaronson, 2023 Workshop on Large Language Models and Transformers, Simons Institute, UC Berkeley\
> > [9] “Robust distortion-free watermarks for language models”, Kuditipudi et al., TMLR 05/2024\
> > [10] “Unbiased watermark for large language models”, Hu et al., ICLR 2024\
> > [11] “Undetectable watermarks for language models”, Christ et al., COLT 2024\
> > [12] “A survey of text watermarking in the era of large language models”, Liu et al., ACM Computing Surveys 2024

---

> > > ### Comment · Reviewer_xcg2 · 2024-11-25
> > >
> > > Thank you for your response!

---

> > > > ### Author Response · Authors · 2024-11-26
> > > >
> > > > We thank the reviewer for acknowledging our rebuttal. We would appreciate if the reviewer could consider raising their score if our rebuttal has addressed all their concerns. We are happy to engage in further discussion otherwise.

---

> > > > > ### Author Response · Authors · 2024-12-01
> > > > >
> > > > > As the discussion period closes soon, we respectfully ask the reviewer to let us know if our rebuttal has addressed all their concerns? If not, we are happy to use the remaining time to clarify outstanding points further.

---

### Official Review · Reviewer_ozah · 2024-11-04

**Soundness:** 3
**Presentation:** 3
**Contribution:** 2
**Rating:** 5
**Confidence:** 3

**Summary:**

This paper explores forgery attacks on LLM watermarks, utilizing traces in forged texts that reflect the forger's partial knowledge, revealing observable differences between genuine watermarked texts and forged watermarked texts. The authors propose a statistical testing method, experimentally verifying that these traces can be detected to identify whether a watermark has been forged.

**Strengths:**

This paper investigates the forgery traces of attacks on LLM watermarks.
It provides definitions through formal descriptions and formula derivations, and designs a statistical testing method. The effectiveness of the testing hypothesis is validated through experiments.
The presentation of figures and tables is relatively clear.

**Weaknesses:**

1.There are various methods for adding watermarks to LLMs, and there may also be multiple ways to forge them. This paper only discusses the red-green watermarking scheme for LLMs and the forgery method of constructing datasets by querying a watermarked model, which is highly restrictive.
2.The rationale for the assumptions in this paper is insufficient. The paper assumes that forgers, when constructing datasets by querying a watermarked model, will leave traces of forgery due to incomplete knowledge in the dataset. However, if the attacker queries the watermarked model again to supplement the missing knowledge in the forged dataset, they can bypass the limitations of this assumption. In this case, the premise does not hold, making the motivation of the paper weak.

**Questions:**

1.The paper should incorporate a comparison of various watermarking methods for LLMs, such as those proposed by Miranda Christ, Sam Gunn, and Or Zamir in "Undetectable watermarks for language models" (COLT, 2024), Rohith Kuditipudi et al. in "Robust distortion-free watermarks for language models" (arXiv, 2023), and Zhengmian Hu et al. in "Unbiased watermark for large language models" (ICLR, 2024). Additionally, it should experimentally compare the effectiveness of the proposed method against the latest forgery techniques, such as those described by Qi Pang et al. in "Attacking LLM watermarks by exploiting their strengths" (arXiv, 2024) and Qilong Wu and Varun Chandrasekaran in "Bypassing LLM watermarks with color-aware substitutions" (arXiv, 2024).
2.Regarding the second Weakness, if an attacker supplements the missing content in the forged dataset by querying the watermarked model again to complete the insufficient knowledge, can this bypass the limitations of the premise assumption and affect the effectiveness of the method proposed in this paper?

---

> ### Author Response · Authors · 2024-11-22
> **Response to Reviewer ozah**
>
> We thank the reviewer for their exhaustive feedback. Below, we address the concerns raised in questions Q1 to Q3; and note that a revised version of the paper has been uploaded, with updates highlighted in blue. We are happy to clarify further if there are additional questions.
>
> **Q1: Why do the authors restrict themselves to learning-based spoofing techniques?**\
> The scope of our work are learning-based spoofers (as defined in Section 2), as they have the ability to generate arbitrary amounts of diverse spoofed text in a cost-effective way and are applicable in realistic setups. This, given the current state of the field, boils down to two methods: Stealing ([1]) and Learnability ([2]). We excluded the spoofing method from [3], as it achieves perfect spoofing on the Unigram scheme [4] (they recover up to 80% of the red-green split and thus can’t be detected) and does not spoof any other schemes. In Section 6, we have already briefly discussed the works of [5] and [6], and why not including them should not be seen as our weakness---we expand on that discussion now.
>
> The scrubbing method presented in [5] could be turned into a somewhat impractical adaptive spoofing attack. Such a potential attack would require a very large computational budget per text generated. Indeed, for each newly generated token in every spoofing attempt, one would have to prompt the watermarked model to perform SCT and pick a green token. We argue that this is not practical in the current form. Moreover, because [5] focuses on scrubbing, no reference implementation nor evaluation of this spoofing attack exists. We believe that designing and evaluating a new practical spoofing attack with ideas from [5] and using it to test our method is outside the scope for our work.
>
> “Piggyback spoofing attacks” ([6]) directly leverage the desirable robustness property of the watermark. Given a watermarked sentence, the attacker modifies a few tokens to alter the meaning of the original sentence while maintaining the watermark, interpreting the result as an instance of spoofing. Mixing human and LM generated text is a realistic use case of LMs, many watermarking methods inherently account for such modifications ([7,9]), making it a priori unclear whether selectively modifying individual tokens is practical spoofing or an inherent property of a watermark. We note that this is also in stark contrast to learning-based spoofers that can generate almost arbitrary text without having to rely on a pre-existing template—which additionally can be both hard to generate for harmful topics and significantly increases costs as it requires queries to the original LM for every generation.
>
> In summary, restricting ourselves to only learning-based spoofers is driven by practical considerations and the state of spoofing attacks. While we do not believe this represents a weakness of our work, we agree this discussion should be visible in the paper—in our new revision we introduce Appendix I that reflects this discussion and motivates our choices.
>
>
> **Q2: Why do the authors restrict themselves to Red-Green watermarks?**\
> We do not see this as a weakness of our work. To be able to evaluate our test, we can only work with schemes where practical spoofing techniques have already been demonstrated. Hence, the only schemes we can study are Red-Green schemes [7], AAR [8] and KTH [9]. As [10, 11] are not considered by any spoofing attacks, these are orthogonal to our work—of course this may change in the future. We include an extended discussion of this in our new Appendix I.
>
> In our main experiment, we focused on Red-Green schemes [7] as they are not only the most popular [12] class of watermarks, but are also seeing increasing adoption by industry (e.g., SynthID recently being announced for Gemini). As the most studied class of schemes, Red-Green also happens to have the most research on spoofing attacks, resulting in several practical attacks [1,2,3]. Combined with the practical adoption of RG schemes, this makes the detection of such spoofed texts a relevant and timely topic of study.
>
> Prompted by the reviewer’s suggestion, as we elaborate in our new Appendix F, the key ideas behind our spoofing detection test generalize to both AAR [8] and KTH [9], where we show that Distillation of both schemes also leaves detectable artifacts. In particular, on AAR, we achieve a TPR at 1% of 90% with 500 tokens of text. On KTH (EXP variant), we achieve a TPR at 1% of 60% for $s=4$ (4 maximum permutations of the key) and 30% for $s=256$ with 1000 tokens of text.
>
> We hope that this discussion and the additional results motivate our focus on Red-Green watermarks and highlight that the core ideas behind our method generalize to other schemes.

---

> > ### Author Response · Authors · 2024-11-22
> >
> > **Q3: Can the proposed detection be bypassed with adaptive spoofing?**\
> > As we concluded in Section 7, adaptive spoofing seems a valuable direction to explore as an attempt to circumvent our detection. However, the concrete steps required to build an adaptive spoofer are unclear as this complex idea was not explored in any prior work. In particular, we remark that before our work, the goal of spoofing attacks was only to minimize the cost, while inducing FPRs in the watermark detector [1, 2]. Hence, prior works did not suspect that artifacts could be leveraged to detect the spoofing attempts. We are the first to highlight and substantiate this concern, which we see as our key impact. In light of our work, future works on watermark spoofing have to adjust their goals to also consider the issue of artifact detectability—we do not believe it is viable for us to extensively explore this issue in our current work, and do not see this as a limitation.
> >
> > [1] “Watermark stealing in large language models”, Jovanovic et al., ICML 2024\
> > [2] “On the learnability of watermarks for language models”, Gu et al., ICLR 2024\
> > [3] “Large Language Model Watermark Stealing With Mixed Integer Programming”, Zhang et al., 2024\
> > [4] “Provable robust watermarking for ai-generated text”, Zhao et al., ICLR 2024\
> > [5] “Bypassing LLM Watermarks with Color-Aware Substitutions”, Wu et al., 2024\
> > [6] “Attacking LLM Watermarks by Exploiting Their Strengths”, Pang et al., 2024\
> > [7] “A watermark for large language models”, Kirchenbauer et al., ICML 2023\
> > [8] “Watermarking of large language models”, Scott Aaronson, 2023 Workshop on Large Language Models and Transformers, Simons Institute, UC Berkeley\
> > [9] “Robust distortion-free watermarks for language models”, Kuditipudi et al., TMLR 05/2024\
> > [10] “Unbiased watermark for large language models”, Hu et al., ICLR 2024\
> > [11] “Undetectable watermarks for language models”, Christ et al., COLT 2024\
> > [12] “A survey of text watermarking in the era of large language models”, Liu et al., ACM Computing Surveys 2024

---

> > > ### Comment · Reviewer_ozah · 2024-11-26
> > > **Official Comment by Reviewer ozah**
> > >
> > > Thank you for your response! I carefully reviewed your reply. However, the authors' explanation for why they restrict themselves to Red-Green watermarks is not very clear, as they do not adequately elaborate on the specific reasons for this choice and its impact on the research results. I also considered the opinions and responses of other reviewers and found that the paper's results restrict its practical applicability to certain scenarios. Therefore, I still maintain my original score.

---

> > > > ### Author Response · Authors · 2024-11-26
> > > >
> > > > We thank the reviewer for engaging in the discussion.
> > > >
> > > > However, we are surprised by the reviewer’s reply. We believed to have elaborated the specific reasons for restricting to the Red-Green watermarks in our rebuttal, as acknowledged by other reviewers. Red-Green watermarks are a very active focus of watermarking research, and in fact the only watermark where we can compare multiple learning-based spoofing techniques. The only two other schemes where there even exists a learning-based spoofer are AAR and KTH, and we have shown in the added Appendix F that our method to discover artifacts generalizes to these schemes. Hence, we now show that on all schemes where learning-based spoofers have been experimentally validated, our method is applicable.
> > > >
> > > > We hope this helps clarify our position further. If not, we kindly ask the reviewer if he could point to more specific issues—e.g. what remains unclear about our initial restriction to Red-Green and are there further experiments that would highlight the impact of our method?

---

> > > > > ### Author Response · Authors · 2024-12-01
> > > > >
> > > > > We thank the reviewer again for replying to our rebuttal. Can the reviewer let us know if our last comment helped clarify the remaining concern?

---

### Author Response · Authors · 2024-11-22
**General Response**

We thank the reviewers for their feedback. We are pleased to see that the reviewers consider our contributions novel and innovative (_xcg2_, _yZgn_, _kEcn_), acknowledge that they are backed by a rigorous mathematical framework (_ozah_, _yZgn_, _KEcn_, _n7sN_) and appreciate our extensive experimental evaluation (_ozah_, _xcg2_, _yZgn_).

We have uploaded an updated version of the manuscript where the new content is marked in blue, and responded to all questions and concerns raised by the reviewers in individual responses below. We are happy to further engage in the discussion in case the reviewers have additional questions.

---

### Author Response · Authors · 2024-11-25
**Discussion period closing soon**

We thank reviewers _yZgn_ and _n7sN_ for the discussion; as the discussion period closes soon, we would like to kindly remind reviewers _ozah_, _xcg2_ and _KEcn_ to acknowledge our rebuttal. We are available to discuss if there are any remaining concerns.

---

### Meta-Review · Area_Chair_niPM · 2024-12-26

**Metareview:**

This submission proposes a forensic investigation of spoofing attacks against text watermarks for language models, which are attacks that spoof the watermark signature, injecting it into text not generated by the watermarked model. The submission shows that interestingly, for a relevant watermarking scheme spoofing attack, these attacks can in turn be detected.

This idea is interesting and the investigation by the authors has merit, but, as the topic of this discussion becomes a bit niche, it would have been good to see a broader discussion and formalization of the security problem of counter-spoofing across watermarking methods (i.e. to my understanding, current theory, such as in Section 4.1 is applicable only to the KGW watermark). The current submission reports interesting findings that could be extended into such a, more general, investigation, elevating it from its current current status that is a bit too close to an instance report to several reviewers.

Due to this concern, and based on only muted support by reviewers, I do not recommend acceptance of this submission for now. I hope the authors are going to re-submit their extended analysis.

**Additional Comments On Reviewer Discussion:**

The authors discuss and resolve a number of minor questions and concerns with the reviewers.

---

### Decision · Program_Chairs · 2025-01-22

Reject